# Universal Self-Adaptive Prompting

**Xingchen Wan**[*,1,2]**, Ruoxi Sun**[1]**, Hootan Nakhost**[1]**, Hanjun Dai**[1]**,**
**Julian Martin Eisenschlos**[1]**, Sercan Ö. Arık**[1]**, Tomas Pfister**[1]
[1]Google [2]University of Oxford

{xingchenw,ruoxis,hootan,hadai,eisenjulian,soarik,tpfister}@google.com

## Abstract

A hallmark of modern large language models (LLMs) is their impressive general zero-shot and few-shot abilities, often elicited through in-context learning (ICL) via prompting. However, while highly coveted and being the most general, zero-shot performances in LLMs are still typically weaker due to the lack of guidance and the difficulty of applying existing automatic prompt design methods in general tasks when ground-truth labels are unavailable. In this study, we address this by presenting Universal Self-Adaptive Prompting (USP), an automatic prompt design approach specifically tailored for zero-shot learning (while compatible with few-shot). Requiring only a small amount of *unlabeled* data and an inference-only LLM, USP is highly versatile: to achieve universal prompting, USP categorizes a possible NLP task into one of the three possible task types and then uses a corresponding selector to select the most suitable queries and zero-shot model-generated responses as *pseudo*-demonstrations, thereby generalizing ICL to the zero-shot setup in a fully automated way. We evaluate USP with PaLM and PaLM 2 models and demonstrate performances that are considerably stronger than standard zero-shot baselines and often comparable to or even superior to few-shot baselines across more than 40 natural language understanding, natural language generation, and reasoning tasks.

## 1 Introduction

The recent advancements in large language models (LLMs) are among the most astonishing breakthroughs in artificial intelligence. The modern, massive attention-based (Vaswani et al., 2017) LLMs not only surpass human and previous models in specific natural language processing tasks, but they have also demonstrated impressive general capabilities (Bubeck et al., 2023). Indeed, thanks to both the scaling of LLM sizes and advances in

---

[*]Work done during internship at Google.

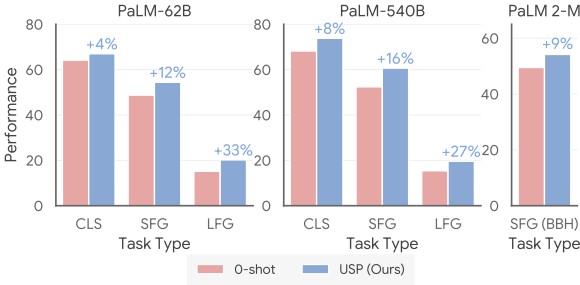

Figure 1: We propose USP, a versatile *zero-shot* prompting method that improves over standard zero-shot prompting across more than 40 Classification (CLS), Short-form Generation (SFG) and Long-form Generation (LFG) tasks (see §3.3 for further explanations in PaLM-62B, PaLM-540B and PaLM 2 models.

training and fine-tuning techniques (Brown et al., 2020; Sanh et al., 2021; Wei et al., 2021), one of the most prominent and impressive abilities of modern LLMs is their *zero-shot* generalizability handling diverse and sophisticated tasks, even if the models have not been explicitly trained on them. Beyond zero-shot abilities, when a few demonstrations are available, the *few-shot* capabilities can take advantage of the information in them with *in-context learning* (ICL) (Brown et al., 2020), leading to further improvements.

Such few-shot capabilities are often observed to improve as the LLMs scale (Brown et al., 2020; Wei et al., 2023). Along with careful prompting, in many cases, LLMs can perform similarly to, or even better than, fine-tuning, even though the latter is both more computationally expensive (due to gradient back-propagation) and more data-intensive. As such, in many scenarios, prompt-based learning has drastically reduced the barrier to the use of even the most massive LLMs.

Notwithstanding the breakthroughs, many open questions remain. While the zero-shot performances of LLMs are highly valued and widely used as a key yardstick of LLM capabilities (Chowdhery et al., 2022; Tay et al., 2022), LLMs still often

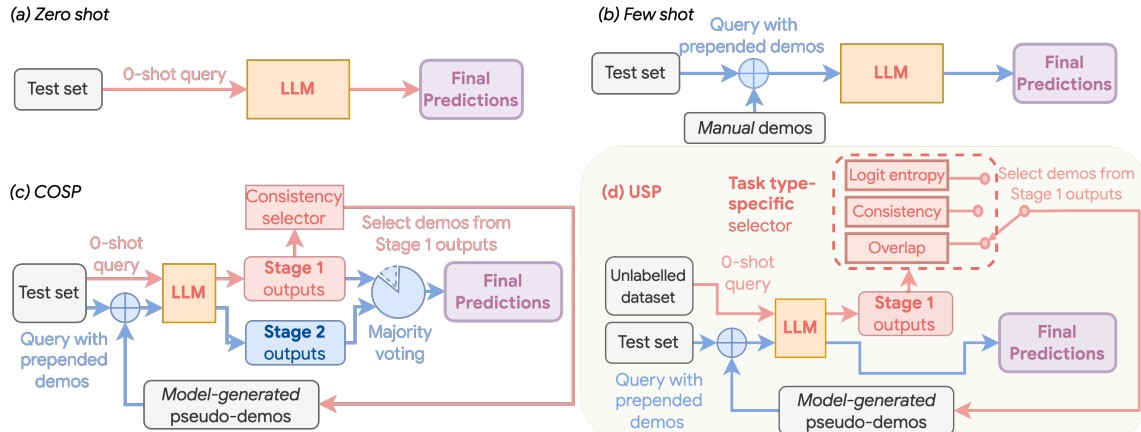

Figure 2: Overview of **(a)** zero-shot setup, **(b)** few-shot setup with in-context learning, **(c)** Consistency-based Self-Adaptive Prompting (Wan et al., 2023) and **(d)** Universal Self-Adaptive Prompting, or USP, the proposed method in this work. The queries *without demos* with which LLMs are directly prompted (zero-shot, or Stage 1 in COSP and USP) are marked in red arrows, and the queries prepended with either the handcrafted demos (few-shot) or model-generated pseudo-demos (Stage 2 in COSP and USP) are marked in blue arrows.

show weaker performances and/or larger performance fluctuations in the zero-shot setting because of the lack of guidance or readily-available template solutions. While many automatic prompting methods have been proposed (refer to §4 for details), few existing works target the zero-shot setup, and heuristic manual prompt design is still often heavily relied upon (Reynolds and McDonell, 2021; Mishra et al., 2022).

On the other hand, even though the ICL paradigm has reduced the cost of data collection and labeling considerably, given that modern LLMs are typically used for an extremely diverse set of tasks, obtaining even a small number of labeled examples per task can easily become expensive for many tasks. Furthermore, in some tasks, obtaining even a few examples might require a non-trivial amount of human effort (e.g., summarization of long articles, translation of low-resource languages, and/or domain-specific question answering requiring research or expertise), or simply impossible for novel tasks that are only revealed at test time.

To address this, we introduce USP (Universal Self-Adaptive Prompting) that specifically pushes the state-of-the-art with ICL in zero-shot settings (while remaining compatible with few-shot) via *pseudo*-demonstrations (pseudo-demos) constructed from *unlabeled* queries and *model-generated* outputs. USP works with fully black-box, inference-only LLMs, and the use of pseudo-demos ensures that USP may operate entirely in the *transductive zero-shot* setup (Xian et al., 2017) using only unlabeled data. This makes USP ex-

tremely versatile, as unlabeled data is typically readily available via, e.g., continuous, on-the-fly collections of user queries. Unlike alternative methods often requiring task knowledge beforehand (e.g., class names), USP requires only the task type information to select an appropriate confidence-quantifying metric (e.g., natural language understanding (NLU) or generation (NLG) – these need to be known anyway), while still remaining capable of using additional information like class names if they are indeed available (§3.3). This enables USP to work in arbitrary, potentially novel tasks at test time and/or tasks that simply cannot be cast as classification problems (e.g., open-domain QA and other generative tasks). USP is inspired by recent works leveraging confident predictions for model self-improvements on chain-of-thought tasks (Wang et al., 2022; Huang et al., 2022; Wan et al., 2023) but inherits the benefits of these works and generalize them considerably in terms of the scope of applicability. To achieve this, we derive various criteria capable of selecting high-quality pseudo-demos in the absence of any ground-truth labels. To summarize:

**1)** We propose USP, a versatile and *black-box* automatic prompting method that can be *zero-shot* using only unlabelled data.

**2)** To achieve this, we select *pseudo-demos* from model-generated outputs via 3 carefully designed scoring functions suitable for different task types.

**3)** As shown in Fig. 1, we show USP realizes large performance gains over more than 40 NLU, NLG and reasoning tasks with PaLM & PaLM 2 models.

## 2 Preliminaries

**In-context Learning (ICL).** ICL enables LLMs to perform few-shot learning by processing several labeled, exemplary queries similar to the test queries we are interested in solving as *demonstrations*, or *demos* in the prompts (Brown et al., 2020; Dong et al., 2022; Logan IV et al., 2022) (Fig. 2b). Formally, denoting a test query as $x$ and if we have $k$ pairs of related concatenated queries and labels $s^{(i)} = \texttt{Concat}(x^{(i)}, y^{(i)}) \, \forall \, i \in \{1, ..., k\}$ serving as demos, we augment the test query by prepending the demos (and instructions, if any) to it:

$$C(x) = \texttt{Concat}(s^{(1)}, ..., s^{(k)}, x). \quad (1)$$

ICL is achieved by obtaining the prediction $\hat{y}$ by querying $C(x)$ instead of just $x$. In our zero-shot setup, *none* of the ground-truth labels (i.e., the $y$s) are available, and we propose to use the LLM predictions themselves as *pseudo*-demos. Thus, our *zero-shot* ICL instead has the form of:

$$\hat{C}(x) = \texttt{Concat}(\hat{s}^{(1)}, ..., \hat{s}^{(k)}, x), \quad (2)$$

where $\hat{s}_i = \texttt{Concat}(x^{(i)}, \hat{y}^{(i)})$, and the ultimate objective of USP is to generate and identify the most suitable set of such pseudo-demos.

**Self-consistency.** For LLMs, Wang et al. (2022) introduce *self-consistency* (SC) for chain-of-thought (CoT) reasoning tasks (Wei et al., 2022b) as an effective approximation of the model confidence – SC decodes each test query multiple times using a non-zero temperature[*]) to introduce stochasticity. The *majority* of the predictions are then chosen as the final predictions.

**COSP.** Inspired by Wang et al. (2022) and entropy minimization (Grandvalet and Bengio, 2004), Wan et al. (2023) propose *Consistency-based Self-Adaptive Prompting* (COSP) to improve zero-shot CoT reasoning. COSP is the most influential prior work to us: as shown in Fig. 2c, COSP uses a two-stage approach. In Stage 1, COSP performs zero-shot inference with multiple decoding paths in a similar manner to SC and then computes the normalized entropy to quantify model confidence via discrepancy in predictions from the same query on different decoding paths. COSP then ranks the Stage 1 outputs based on the entropy (and other metrics such as diversity and repetition) and selects the confident outputs as the pseudo-demos. In

---

[*]We use a temperature of 0.7 following previous works like Wan et al. (2023) and Wang et al. (2022).

Stage 2, these pseudo-demos are prepended to the test queries in a manner similar to few-shot ICL, and the final predictions are given by the majority vote over outputs in both stages.

## 3 Universal Self-Adaptive Prompting

### 3.1 Motivation and Challenges of USP

Inspired by the success of COSP, we argue that the principle of confidence-based prompting should be *universally* applicable to *all* tasks, rather than being exclusive to a narrow set of reasoning tasks COSP considered; this forms the motivation and the goal of this paper. However, a number of limitations and challenges prohibit a trivial generalization: first, a universal prompting strategy needs to accommodate numerous, vastly diverse tasks that vary significantly in terms of objective, prompting, evaluation, and, unsurprisingly, confidence/uncertainty quantification. As a result, SC and the techniques developed by Wan et al. (2023) may be sub-optimal or even inapplicable for other task types: for instance, many problems are cast as classification where the output well-calibrated logits are useful for uncertainty quantification, but such information is not used in the original formulation of COSP. Also, the notion of majority voting crucial to COSP and SC may not even exist for creative and generative tasks with many plausible solutions.

### 3.2 Overview of USP

To address the challenges, we present USP (Fig. 2d and Algorithm 1). USP shares some high-level similarities to the COSP formulation: USP also adopts a two-staged approach where in Stage 1, the LLMs are prompted in a zero-shot manner to generate a collection of candidate responses from which a few *model-generated* pseudo-demos are selected; in Stage 2, USP prepends these pseudo-demos to the test queries in a few-shot manner (Eq. (2)) and prompts the LLM again to obtain the final predictions. However, we highlight a few key design decisions, in particular those differing from COSP, that effectively overcome the aforementioned challenges and enable USP to generalize:
**1.** *Task-specific pseudo-demo selector.* The pseudo-demo selector, which selects the most suitable query-response pair from the zero-shot outputs, is central to USP. With reference to Fig. 2c and 2d, whereas COSP only uses the consistency-based selector and hence is only applicable to a limited number of tasks, USP instead uses a *task-type spe-*

**Algorithm 1** USP. Stage 1 steps are marked in red, and Stage 2 steps are marked in blue.

---

1: **Input**: Test set with size $N$: $\mathcal{T} = \{x^{(i)}\}_{i=1}^{N}$, unlabeled set for demo generation with size $N_u$: $\mathcal{D} = \{d^{(j)}\}_{j=1}^{N_u}$ (can be same as or a subset of $\mathcal{T}$, or a different but related set of unlabeled queries), Pool of generated responses $\mathcal{P} \leftarrow \emptyset$, Task type $t \in \{\texttt{CLS}, \texttt{SFG}, \texttt{LFG}\}$ (§3.3).
2: **Output**: Predictions $\{\hat{y}^{(i)}\}_{i=1}^{N}$.
3: **for** $j \in [1, N_u]$ **do**
4:    [Stage 1] Query the LLM with $d^{(j)}$ under the zero-shot setup to obtain a *single* prediction $\hat{z}^{(j)}$ (*if* $t$=CLS), or query $m$ times with non-zero temperature to obtain $m$ predictions $\{\hat{z}_k^{(j)}\}_{k=1}^{m}$ (*otherwise*).
5:    Add eligible candidate pseudo-demos $\{p_j\}_{j=1}^{N_u}$ (from concatenating $d^{(j)}$ and $\hat{z}^{(j)}$) to $\mathcal{P}$.
6: **end for**
7: Build the pseudo-demo set $\mathcal{S} = \{s_1, ..., s_K\}$ (with $|\mathcal{S}| = K$) from $\mathcal{P}$ with one of the selectors in §3.3 depending on $t$.
8: **for** $i \in [1, N]$ **do**
9:    [Stage 2] Concatenate the $\mathcal{S}$ to $x^{(i)}$ (Eq. 2) and query again (with greedy decoding for generative (SFG/LFG) tasks) to obtain the final LLM prediction $\hat{y}^{(i)}$.
10: **end for**

---

*cific* selector that is key for its versatility – we explain this in detail in §3.3.

**2.** *Separating test set and the demo-generating dataset.* Instead of expecting the *full* test set $\mathcal{T}$ in Stage 1, USP expects a general unlabeled dataset $\mathcal{D}$, which can be the full test set $\mathcal{T}$, a subset of it, a different unlabelled set, or possibly even a model-generated dataset like Schick and Schütze (2021) (although we always use a subset of $\mathcal{D}$ for simplicity in this work). Its sole purpose is to generate the pseudo-demos, enabling USP to work even if $\mathcal{T}$ is not known a-priori in its entirety. Indeed, as we will show in §5, USP is capable of generating high-quality pseudo-demos with *only 64 unlabeled samples* per dataset. This makes USP more *sample efficient*, due to the smaller number of unlabeled samples required, and more *computationally efficient*, as the algorithm only needs to iterate through $\mathcal{D}$, which can be modestly sized, in Stage 1.

**3.** *Dropping reliance on majority vote.* The use of majority vote (as shown in Fig. 2c) is crucial for COSP, but as discussed, the procedure is also computationally expensive and inapplicable when the majority itself is ill-defined. To address this, by default, USP instead only decodes *once* in Stage 2 with *greedy decoding* (i.e., temperature $= 0$) and uses the maximum likelihood estimated (MLE) outputs as the final predictions. It is worth noting that USP remains compatible with majority voting over multiple decoding (if it can be used) for further performance improvements, but no longer *depends*

*on* these to function.

### 3.3 Task-specific Selector

The objective of the *selector* (Step 7 in Algorithm 1) is **1)** to build a pool of candidate pseudo-demos $\mathcal{P}$, whose elements $p^{(j)}$ are formed concatenating dataset queries $\{d^{(j)}\}_{j=1}^{N_u}$ and their zero-shot LLM predictions $\{\hat{z}^{(j)}\}_{j=1}^{N_u}$ and **2)** to select $\mathcal{S}$, a subset of $K$ pseudo-demos from $\mathcal{P}$ to be prepended to the test queries. We use a function $\mathcal{F} : \mathcal{P} \to \mathbb{R}$ (the design of $\mathcal{F}$ is explained later in this section) to "score" each candidate. We select the first pseudo-demo in $\mathcal{S}$ by finding the maximizer of $\mathcal{F}(\cdot)$ in $\mathcal{P}$. For each of the subsequent pseudo-demos $k \in \{2, ..., K\}$, we instead repeatedly find the maximizer of $\mathcal{F}(\cdot)$ with a diversity-promoting term to penalize candidates that are too similar to *any* of the pseudo-demos already selected and add to $\mathcal{S}$:

$$s_k = \operatorname*{arg\,max}_{p \in \mathcal{P} \setminus \mathcal{S}_{1:k-1}} \mathcal{F}(p) - \lambda \max_{k'=1}^{k-1} \Big( S_c\big(\phi(p), \phi(s_{k'})\big)\Big), \quad (3)$$

where we follow Wan et al. (2023) to set $\lambda$, the trade-off parameter, to 0.2 in all experiments without further tuning and use $z$-score standardization for the two terms in Eq. (3) over $\mathcal{P}$ to ensure they are of a comparable magnitude; $S_c(\cdot, \cdot)$ denotes the cosine similarity and $\phi(\cdot)$ is the sentence-level embedding given by an auxiliary model, as in COSP. The design of $\mathcal{F}(\cdot)$, therefore, encodes our preference on which pseudo-demos should be prepended to the test queries for ICL. To achieve *universal* prompting, we categorize a possible task into one of the three generic types in Table 1. We use this categorization to design task-specific scoring functions $\mathcal{F}(\cdot)$ below, and empirically validate the effectiveness of these designs in §5.

| Task type | # possible responses | # correct responses | Logits required? | Score fn. |
|-----------|---------------------|---------------------|------------------|-----------|
| CLS | Few | Single | Yes | Eq. (4) |
| SFG | Many | Single/few | No | Eq. (7) |
| LFG | Many | Many | No | Eq. (8) |

Table 1: Categorization of the NLP tasks in USP, namely Classification (CLS), Short-form Generation (SFG) and Long-form Generation (LFG).

**Classification (CLS).** With reference to Table 1, we first consider problems that feature the selection of a single correct answer from a few possible options – we use the descriptor CLS for "classification", as the label space $\mathcal{C}$ in this case is small

and known, and the task is to pick the most probable class $\mathcal{C}$: $\hat{z}^{(j)} = \arg\max_{c \in \mathcal{C}} \mathbb{P}(c|d^{(j)})$. Since the logits are available in this case, we do *not* need self-consistency to estimate the prediction confidence, although we may still choose to use a self-consistency-based confidence metric if, the model would be poorly calibrated with logits, or self-consistency would be preferable due to other reasons (e.g., when CoT prompting is used and generating diverse reasoning paths via multiple-path decoding is beneficial – see the next paragraph on SFG for details). Instead, for $p^{(j)} =$ Concat$(d^{(j)}, \hat{z}^{(j)}) \in \mathcal{P}$, we simply query the LLM once and use the negative entropy of the distribution over $\mathcal{C}$ as the function $\mathcal{F}$ for the CLS case:

$$\mathcal{F}_{\mathsf{CLS}}(p^{(j)}|d^{(j)}) := \sum_{c \in \mathcal{C}} \tilde{\mathbb{P}}(c|d^{(j)}) \log \tilde{\mathbb{P}}(c|d^{(j)}), \quad (4)$$

where $\tilde{\mathbb{P}}(c|d^{(j)})$ is the normalized probability with $\sum_{c \in \mathcal{C}} \tilde{\mathbb{P}}(c|d^{(j)}) = 1$ – it is worth noting that orthogonally, an improved uncertainty metric like the *semantic uncertainty* (Kuhn et al., 2023)) may be used instead, although we do not consider these in the present work. We further use the knowledge of $\mathcal{C}$ to ensure good coverage of the label space, which has been shown to be important for a strong ICL performance (Min et al., 2022). Specifically, to build $\mathcal{S}$, instead of simply generating $K$ pseudo-demos from $\mathcal{P}$, we generate $K/|\mathcal{C}|$ pseudo-demos *for each $c \in \mathcal{C}$ from a subset $\mathcal{P}_c \subset \mathcal{P}$* where:

$$\mathcal{P}_c = \left\{ p^{(j)} \in \mathcal{P} \text{ if } \hat{z}^{(j)} = c \,\forall j \in \{1,...,N_u\} \right\}. \quad (5)$$

This is because LLMs can be more confident in some classes, and simply choosing the most confident predictions overall as pseudo-demos may lead to bias towards these classes; we mitigate this to ensure that the selected pseudo-demos $K$ feature each class approximately uniformly. Note that it is possible that $K < |\mathcal{C}|$ or $\mathrm{mod}(K, |\mathcal{C}|) \neq 0$. In these cases, we generate $\lceil \frac{K}{|\mathcal{C}|} \rceil$ pseudo-demos *per class* and prepend each test query $x^{(i)} \in \mathcal{T}$ with $K$ randomly sampled pseudo-demos to ensure fairness *in expectation* over $\mathcal{T}$. Lastly, it is possible that some classes are never predicted in $\mathcal{D}$, e.g., an over-confident model may never predict the "*not sure*" option in inference tasks. As a result, the set $\mathcal{P}_c$ in Eq. (5) is empty for these unpredicted classes. To nevertheless generate the most plausible pseudo-demos for them, for an unpredicted class $c_u$, we pick the top queries in $\mathcal{D}$ with the highest model-assigned probability in $c_u$:

$$\mathrm{Top}\frac{K}{|\mathcal{C}|}\Big|_{d^{(j)} \in \mathcal{D}} \Big( \mathbb{P}(c = c_u|d^{(j)}) \Big), \quad (6)$$

noting that the indexing is over the unlabeled dataset $\mathcal{D}$. These queries are then concatenated with class label $c_u$ to form the pseudo-demos for these unpredicted classes.

**Short-form Generation (SFG).** We use descriptor SFG (for *Short-form Generation*) to denote the class of generation problems typically with many possible responses but only one to a few correct responses, and examples include *Question Answering*. Alternatively, as we discussed in the previous paragraph, we may use the SFG formulation for CLS tasks if we use the text-to-text formulation like T5 (Raffel et al., 2020), have no access or prefer not to rely on logits, or as discussed when self-consistency-style multiple decoding is preferable. Unlike the CLS case, we assume access to only the model outputs $\hat{z}^{(j)}$ but not the logit distribution. This covers the case covered in COSP (problems such as arithmetic reasoning considered in COSP fall into this category), and thus we may use the *normalized entropy* in Wan et al. (2023) to gauge the model confidence, except that for non-CoT prompted tasks, we skip the rationale generation step and prompt for answers directly. Specifically, for each $d^{(j)} \in \mathcal{D}$, we query the LLM $m$ repetitions, under temperature sampling to obtain $m$ predictions $\{\hat{z}_\ell^{(j)}\}_{\ell=1}^m$. While only the *majority* predictions of each query are added to $\mathcal{P} := \left\{ \mathsf{Maj}\big( \{\hat{z}_\ell^{(j)}\}_{\ell=1}^m \big) \right\}_{j=1}^{N_u}$, we use all $m$ predictions to score the model confidence for each $p^{(j)} \in \mathcal{P}$:

$$\mathcal{F}_{\mathsf{SFG}}\big(p^{(j)}\big|\{\hat{z}_\ell^{(j)}\}_{\ell=1}^m\big) := -\frac{\sum_{\alpha=1}^\mu \tilde{\mathbb{P}}(\hat{z}_\alpha^{(j)}) \log \tilde{\mathbb{P}}(\hat{z}_\alpha^{(j)})}{\log m}, \quad (7)$$

where $\mu \leq m$ is the number of *unique* answers and $\tilde{\mathbb{P}}(\hat{z}_\alpha^{(j)})$ is the empirical frequency of an *unique* answer $\hat{z}_\alpha^{(j)}$ in all $m$ predictions for $d^{(j)}$.

**Long-form Generation (LFG)** The final category, LFG for *Long-form Generation*, features NLG tasks with longer responses and many plausible responses with typical examples being summarization and translation. As discussed, Eq. (7) does not effectively approximate confidence/uncertainty in this case, as decoding the same query with temperature sampling $m$ times is unlikely to yield identical responses in terms of surface texts due to the length

of generation, *even for the confident predictions*. On the other hand, it would also be challenging to apply logit-based modeling in the face of the high-dimensional joint probabilities & the presence of sequential relationships amongst the generated tokens. To measure confidence in this case, we first follow the SFG case by querying each $d^{(j)} \in \mathcal{D}$ for $m$ repetitions $\{\hat{z}_\ell^{(j)}\}_{\ell=1}^m$. Instead of using Eq. (7), we compute the *average pairwise* ROUGE score between all pairs of the $m$ responses:

$$\mathcal{F}_{\mathsf{LFG}}\big(p^{(j)}\big|\{\hat{z}_\ell^{(j)}\}_{\ell=1}^m\big) := \frac{2\sum_{\substack{\ell=1,\ell'=1 \\ \ell'\neq\ell}}^m \mathsf{ROUGE}(\hat{z}_\ell^{(j)}, \hat{z}_{\ell'}^{(j)})}{m(m-1)}, \tag{8}$$

where another overlap metric, such as the pairwise BLEU (Shen et al., 2019) or the sentence-level embedding cosine similarity from an auxiliary model, may be used instead. Another challenge for LFG tasks is that unlike SFG where $\mathcal{P}$ can be simply built from majority predictions for each query $d^{(j)} \in \mathcal{D}$, "majority" is no longer well-defined. We thus use $\mathcal{F}_{\mathsf{LFG}}$ to rank the confidence of the queries in $\mathcal{D}$ & determine which *queries* to be used in $\mathcal{S}$ *only*. For the *response* part of the pseudo-demos, we decode the LLM again *with argmax decoding* to obtain the MLE predictions on the selected queries to build $\mathcal{S}$. Lastly, given that zero-shot text generation is purely driven by prompting and instructions, we observe that the LLMs sometimes generate extremely confident text completions instead of actually completing the instructed tasks (e.g., summarization); selecting these outputs as pseudo-demos, as we investigate in §5, can significantly degrade performance. Given that these outputs often feature an abnormally high $\mathcal{F}_{\mathsf{LFG}}$ score, we apply a simple but canonical outlier filtering technique to remove queries with score > upper quartile + 1.5×interquartile range (IQR) (Tukey et al., 1977).

### 3.4 Cost Analysis

Computing the USP scores itself is cheap, and the cost is thus bottlenecked by the amount of processing from the LLM side. In particular, the additional costs are:

• *Stage 1*: with $|\mathcal{D}|$ unlabeled samples, we require $|\mathcal{D}|$ additional model queries for the CLS task and $64m$ (we use $m = 6$) for SFG and LFG tasks – it is worth noting that we can also use batching to parallelize this step. As seen in Table 5 in App. B, the column $|\mathcal{D}|/|\mathcal{T}|$ represents the fraction of the unlabeled samples to the size of the entire test set, the additional cost is always negligible compared

to the cost we need to incur anyway by iterating over the test set, except for some very small-scale toy tasks with small test tasks.

• *Stage 2*: This stage is completely identical to standard few-shot in-context learning.

Thus, compared to standard zero-shot learning, USP requires the additional Stage 1, which typically only adds a small amount of cost, as discussed above. In Stage 2, the LLM needs to process a longer context due to the use of pseudo-demos for in-context learning. However, this is due to the use of in-context learning and is not an additional cost uniquely attributable to USP – it is true for all other methods relying on ICL. Compared to few-shot learning, the only additional overhead is the use of Stage 1, but crucially, no labeled data is required at any point in time.

## 4 Related Works

Besides those covered in §2, here we discuss other related works in zero-shot automatic prompting. We include an additional literature review in the App. A.

AutoCoT (Zhang et al., 2022) also uses model-generated output as pseudo-demos but differs in the selection procedure – it computes a sentence embedding of available queries and uses clustering to select the centroid queries as pseudo-demos. This process, unlike USP, is purely based on the query (dis)similarity rather than the output quality, and the quality of the selected pseudo-demos is thus, in expectation, the same as the average model performance – we empirically compare against a generalized version of it in §5, which is originally designed for reasoning tasks only (hence the name).

Another method, Z-ICL (Lyu et al., 2022), generates pseudo-demos with synonyms of random class names. It, however, by assuming label knowledge, is limited to a subset of CLS tasks where it is reasonable to do label synonym replacement. For example, while it is reasonable to replace simple sentiment-describing labels like "good" or "bad", the same may not be possible for factual labels or when labels are beyond single words (e.g., the race{h,m} examples shown in Table 9). Randomly selecting labels also only generates correct demos with a probability of $\frac{1}{|\mathcal{C}|}$ – given the recent discovery that modern LLMs genuinely learn from the demos and can be sensitive to their correctness (Wei et al., 2023), providing mostly wrong demos is sub-optimal. To represent this class of methods,

we compare against a *Random demo* baseline in our experiments (see §5 for details).

| Setting | Zero-shot | | | | Few-shot |
|---|---|---|---|---|---|
| Method | 0-shot | Auto-CoT | Random demo | USP (*Ours*) | 5-shot |
| winogrande | 80.51 | 83.98 | **85.56** | 85.48 | 80.58 |
| piqa | 81.50 | 83.19 | **84.28** | 83.13 | 83.84 |
| storycloze | 82.10 | 81.40 | 83.54 | **85.84** | 86.26 |
| anlir1 | 48.60 | 54.10 | 53.60 | **58.50** | 58.30 |
| anlir2 | 43.70 | 52.00 | 50.70 | **54.00** | 53.00 |
| anlir3 | 46.25 | 55.58 | 55.33 | **59.67** | 56.67 |
| boolq | 87.77 | 89.66 | 90.15 | **90.18** | 89.08 |
| copa | 93.00 | 95.00 | **97.00** | 94.00 | 96.00 |
| rte | 72.56 | 80.51 | **81.23** | 79.78 | 80.87 |
| wic | 57.52 | 56.90 | 57.37 | 57.37 | **62.70** |
| wsc | 88.42 | 88.42 | 87.37 | **89.47** | 83.51 |
| arc_e | 78.77 | 87.02 | 85.96 | **88.16** | 87.32 |
| arc_c | 50.64 | **60.60** | 56.39 | 60.17 | **61.80** |
| raceh* | 45.88 | **50.20** | 49.23 | **50.57** | 50.00 |
| racem* | 65.95 | **69.78** | 69.29 | **70.61** | 69.29 |
| *Average ↑* | *68.21* | *72.56* | *72.47* | ***73.80*** | *73.28* |
| *Gain over 0-shot (%)* | *0.00* | *6.37* | *6.24* | ***8.19*** | *7.43* |
| *Average rank ↓* | *4.53* | *3.00* | *2.87* | ***2.00*** | *2.40* |

Table 2: **Accuracy** on CLS tasks (Table 1) with PaLM-540B (Chowdhery et al., 2022) (Refer to App. D.1 for results with PaLM-62B). *Methods in the **Zero-shot** columns do not use ground-truth label guidance and generate 5 pseudo-demos if applicable, whereas the **5-shot** results use 5 human-labeled in-context demos.* The top two results for each model are bolded and ranked by color: **best** and **second-best**. ↑: larger is better. ↓: smaller is better. *See notes in App. C.1.

# 5 Experiments

**Setup.** On PaLM-540B and PaLM-62B (Chowdhery et al., 2022), we consider a wide variety of CLS, SFG and LFG tasks, and the readers are referred to App. B for more details. We also experiment on the state-of-the-art PaLM 2-M (Google et al., 2023) model and test it on BIG-bench Hard (BBH) tasks, a suite of challenging tasks often requiring complicated reasoning, logic, or manipulations where previous models underperform humans (Suzgun et al., 2022). We compare USP against (i) standard zero-shot prompting (except for BBH tasks where we use standard zero-shot-*CoT* prompting (Kojima et al., 2022) (*0-shot*); (ii) an adapted version of AutoCoT (Zhang et al., 2022) for general NLP tasks (*AutoCoT*); (iii) *Random demo*, where we follow all of the USP procedure except we randomly sample $K$ demos from $\mathcal{P}$ – this serves both as an ablation baseline to USP and as a generalization for methods like Z-ICL described in §4 which only work for CLS tasks, except that *Random demo* is arguably stronger as it samples from the *model predictions* rather than *possible classes*, the former of which is more likely to yield correct pseudo-

demos as long as the LLM is better than random guessing in zero shot; (iv) standard few-shot with golden demonstrations (*k-shot* where $k$ depends on tasks; see explanation in result tables). For a fair comparison, *AutoCoT*, *Random demo* and USP all generate $k$ pseudo-demos per sample from 64 randomly sampled, unlabelled test queries per task (i.e., $\mathcal{D}$ in §3.3). We include all other implementation details in App. C.

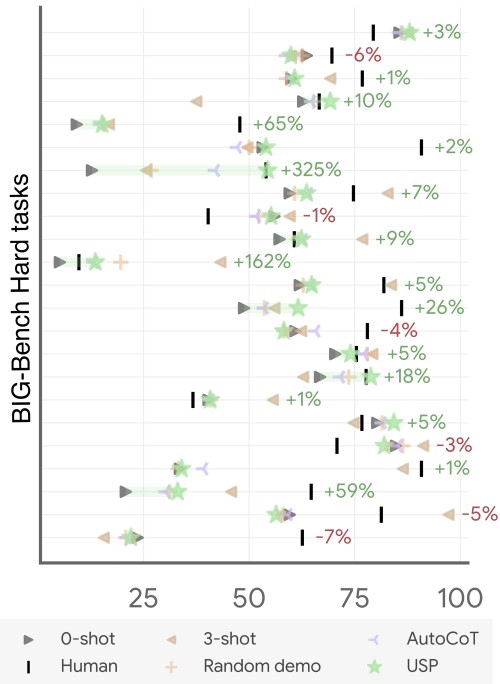

| Setting | Zero-shot | | | | Few-shot |
|---|---|---|---|---|---|
| Method | 0-shot | Auto-CoT | Random demo | USP (*Ours*) | 3-shot |
| *Average ↑* | *49.50* | *52.56* | *52.01* | ***54.18*** | ***60.39*** |
| *Gain over 0-shot (%)* | *0.00* | *6.19* | *5.07* | ***9.45*** | ***21.99*** |
| *Average rank ↓* | *3.79* | *3.04* | *3.18* | ***2.50*** | ***2.14*** |

Figure 3: **Accuracy** on BIG-Bench Hard tasks with PaLM 2-M (each line represents a task of the suite – refer to App. B for full details). The gain/loss of USP over standard 0-shot is shown in percentages. Note that 3 (pseudo-)demos are generated per query following Google et al. (2023). *Human* refers to average human performance from Suzgun et al. (2022).

**Discussion of main results.** We show the results of CLS, SFG and LFG tasks with PaLM-540B in Tables 2, 3 and 4, respectively, and BBH results on PaLM 2-M are shown in Fig. 3 (examples of the generated pseudo-demos in representative tasks are shown in Table 12 and 13 in App. D.4 and PaLM-62B results are shown in App. D.1). We find that USP greatly improves upon standard zero-shot prompting without any pseudo-demos, out-

| Setting | Zero-shot | | | | Few-shot |
| Method | 0-shot | Auto-CoT | Random demo | USP *(Ours)* | 5-shot |
|---|---|---|---|---|---|
| lambda[a] | **78.71** / - | 77.70 / - | 76.13 / - | 75.01 / - | **77.91** / - |
| web_questions | 10.33 / 23.60 | 16.04 / 31.76 | 20.47 / 36.55 | **25.64 / 43.31** | **33.61 / 47.92** |
| natural_questions | 20.49 / 31.04 | 29.31 / 39.36 | 29.00 / 39.34 | **32.19 / 43.56** | **35.88 / 46.50** |
| triviaqa_wiki | 76.73 / 81.85 | 78.73 / 84.05 | **80.52 / 84.89** | 80.10 / 84.57 | 73.78 / 79.52 |
| squad* | 75.67 / 80.85 | **90.93 / 94.37** | 88.47 / 92.93 | 90.29 / 94.06 | 88.83 / 92.39 |
| *Average* ↑ | *52.39 / 59.21[b]* | *58.54 / 65.45[b]* | *58.92 / 65.97[b]* | *60.64 / 68.10[b]* | ***62.00 / 68.85[b]*** |
| *Gain over 0-shot (%)* | *0.00 / 0.00[b]* | *11.75 / 10.54[b]* | *12.47 / 11.41[b]* | *15.76 / 15.02[b]* | ***18.36 / 16.27[b]*** |
| *Average rank* ↓[c] | *4.00* | *2.80* | *3.20* | *2.60* | ***2.40*** |

Table 3: **Exact Match (EM) / F1** on SFG **tasks** with PaLM-540B (Refer to App. D.1 for results with PaLM-62B). [a]Only EM shown as lambda expects a single correct answer. [b]Used lambda EM for the average F1 score. [c]Ranked in terms of EM. *See notes in App. C.1. Refer to Table 2 for further explanations.

| Setting | Zero-shot | | | | Few-shot |
| Method | 0-shot | Auto-CoT | Random demo | USP *(Ours)* | 1-shot |
|---|---|---|---|---|---|
| xsum | 18.4 / 14.7 / 0.186 | **20.5 / 15.3 / 0.347** | 18.0 / 14.1 / 0.301 | 19.3 / 14.9 / 0.329 | **23.6 / 18.6 / 0.337** |
| wikilingua (en) | 20.1 / 16.1 / 0.390 | 14.1 / 11.6 / 0.399 | 21.2 / 17.2 / 0.425 | **30.5 / 24.3 / 0.496** | 29.7 / 24.0 / 0.488 |
| *Average* ↑ | *19.3 / 15.4 / 0.288* | *17.3 / 13.4 / 0.373* | *19.6 / 15.6 / 0.363* | *24.9 / 19.6 / 0.412* | ***26.7 / 21.3 / 0.413*** |
| *Gain over 0-shot (%)* | *0.0 / 0.0 / 0.0* | *-10.3 / -12.6 / 29.8* | *1.7 / 1.8 / 26.3* | *29.2 / 27.4 / 43.3* | ***38.3 / 38.5 / 43.4*** |

Table 4: **ROUGE-1 / ROUGE-Lsum / BLEURT (Sellam et al., 2020) scores** on LFG **tasks** with PaLM-540B (Refer to App. D.1 for results with PaLM-62B). Note that due to the longer context length in LFG problems considered, we generate 1 pseudo-demo under zero-shot setting (if applicable), and use 1 demonstration under few-shot setting (instead of 5 in Tables 2 and 3). Refer to Table 2 for further explanations.

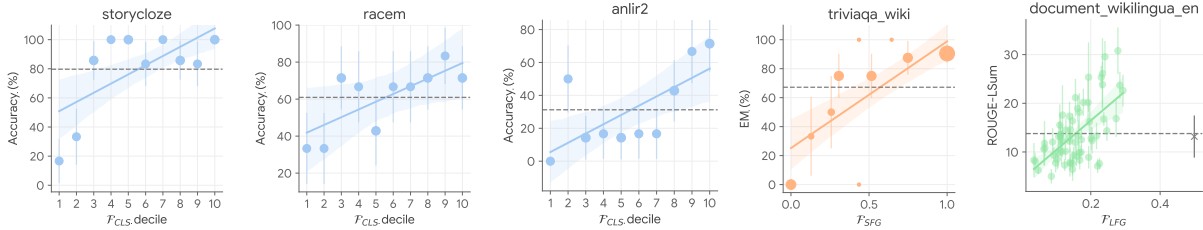

Figure 4: *USP picks confident predictions that are more likely better.* Ground-truth performance metrics in the Stage 1 unlabelled samples ($\mathcal{D}$) against USP scores in selected tasks with PaLM-540B: $\mathcal{F}_{CLS}$ against accuracy (CLS), $\mathcal{F}_{SFG}$ against EM (SFG), and $\mathcal{F}_{CLS}$ against ROUGE-LSum (LFG). CLS: single-sample accuracy is binary and we discretize $\mathcal{F}_{CLS}$ into 10 deciles & show the mean acc. $\pm$ 1 SEM in each bin. SFG: Same as CLS, except that $\mathcal{F}_{SFG}$ is already discrete & no further discretization is performed; marker sizes are proportional to numbers of samples of each $\mathcal{F}_{SFG}$ value. LFG: Both the evaluation metric and $\mathcal{F}_{LFG}$ are continuous and we plot all data without aggregation – since we query each $d^{(j)} \in \mathcal{D}$ 6 times, we show the mean $\pm$ SEM ground-truth ROUGE score for each $d^{(j)}$; gray $\times$ markers denote outliers. The overall mean performance over $\mathcal{D}$ (gray dashed lines) and linear trend lines & confidence intervals are shown in all plots. More results are provided in the App. D.3.

performs other zero-shot methods using pseudo-demos, and is often competitive to or better than few-shot prompting, all achieved with only 64 un-labeled samples per task. Generally, we find the gain margin to be larger in generative tasks and in larger and/or more advanced models. We hypothesize that 1) LLMs benefit more on guidance from the demonstration in generative tasks, which essentially feature unbounded action spaces, whereas in CLS, the LLM only needs to select a response out of a few; 2) larger models and/or those trained with more advanced techniques (e.g., instruction fine-tuning) have stronger ICL capabilities to take advantage of the demos of better quality.

**Few-shot USP.** On the BBH tasks on PaLM 2, we also test a *few-shot* variant of USP (termed USPf) to generate additional pseudo-demos on top of scarce, manual demos. We show the results in Fig. 9 in App. D, and USPf outperforms both the zero-shot USP reported in Fig. 3 and standard

1-shot, thereby highlighting the generality of USP.

***How* does USP work?** To analyze how the USP procedure (§3.3) improves performance, we plot the USP scores against the *ground-truth performance* (accuracy, EM or ROUGE) of the queries in unlabeled datasets $\mathcal{D}$ (with $|\mathcal{D}| = 64$) in Fig. 4 (additional results are reported in App. D), and we observe that across task types and difficulty levels (as measured by the average performance marked by the gray dashed lines in Fig. 4), the USP scores are generally well-correlated with the ground-truth performance, which also validates the finding that LLMs "mostly know what they know" (Kadavath et al., 2022). The recent findings that larger LLMs genuinely learn information from in-context examples (instead of simply following a prompt format) and thus benefit more from correct examples (Wei et al., 2023) are consistent with the results of USP, which, as we show, is more likely to generate correct/high-quality pseudo-demos. Interestingly, a concurrent work (Margatina et al., 2023) also shows that *even when golden labeled examples are available*, better in-context examples still tend to exhibit low uncertainty and diversity.

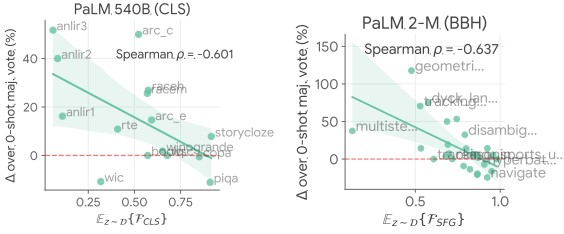

Figure 5: *Gain from USP is larger with higher zero-shot uncertainty.* Relative gain of Stage 2 over Stage 1 accuracy/EM in PaLM-540B/CLS tasks (**left**) & PaLM 2-M/BBH tasks (**right**) against *average USP score*: $\mathbb{E}_{z \sim \mathcal{D}}[\mathcal{F}_{\text{CLS/SFG}}(z)]$. A *higher* average USP score denotes *lower* zero-shot uncertainty. Trend lines and confidence intervals (shades) are shown.

***When* does USP work better?** While USP improves generally, there are cases where USP underperforms standard zero-shot – this seemingly counter-intuitive phenomenon is not unique to USP and is common even for few-shot learning with golden examples from both our results and previous works (Brown et al., 2020; Chowdhery et al., 2022). Nonetheless, understanding *when* it happens for specific tasks can be crucial for users' decision-making. As shown in Fig. 5, we find the *average Stage 1 USP score across $\mathcal{D}$* to be a good *zero-shot*

indicator of the extent of improvement from USP. An intuitive explanation is the average USP score quantifies the general uncertainty the model has about the task (and potentially the task difficulty): with a high average USP score, the model is already confident under zero-shot, and the benefits from ICL are lower (and sometimes may even worsen performance). On the other hand, a low average USP score suggests high model uncertainty and larger potential gains from additional guidance.

# 6 Conclusion

We propose USP, a versatile, *zero-shot* automatic prompting technique applicable to a wide range of NLU, NLG, and reasoning tasks. We show large improvement over standard zero-shot prompting and other baselines in over 40 tasks with 3 LLMs.

## Limitations

We believe that the room for future improvements is ample:

First, the present work specifically targets in-context demonstrations, a sub-component of the overall prompt, and it does not attempt to optimize the other components; a future work would be relaxing this restriction and combining USP with orthogonal techniques (e.g., calibration methods (Zhao et al., 2021; Han et al., 2023; Zhou et al., 2023a) and black-box methods targeting other parts of the overall prompt (Deng et al., 2022; Zhou et al., 2023b)) for improved flexibility.

Second, while our method is general in terms of the *tasks*, it might be more demanding on the *model capabilities*: for the USP score to function as intended, we implicitly demand the model to generate well-calibrated outputs in terms of uncertainty, and the ICL formulation also requires strong in-context learning abilities, both of which have been shown to correlate strongly with model sizes (Kadavath et al., 2022; Wei et al., 2022a). Third, the present work only considers tasks with natural language outputs. Given the ever-improving capabilities of LLMs, it would also be interesting to apply the idea in more novel setups, including but not limited to planning (where LLMs act as autonomous, environment-interacting agents) and multi-modal settings beyond pure NLP problems.

Lastly, we note that especially for the generative tasks (SFG and LFG), in many cases USP greatly improves the zero-shot performance but does not always completely close the gap compared to the few-

shot baseline using golden examples. There are also cases where USP does not meaningfully improve over zero-shot baselines. While we provide a brief analysis in §5 to investigate when that happens, it would also be helpful to investigate whether there are potential remedies, especially given that, as discussed, such occasional performance deterioration even occurs with few-shot prompting with golden demonstrations. We defer thorough investigations to future work.

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

# A  Additional Related Works

In this section, we discuss additional prior works that are related to USP in various aspects.

**Bootstrapping LLM knowledge.**  The promising abilities of LLMs have led to efforts to improve them with their own outputs: Meng et al. (2020) use class names only and self-training to improve text classification; Zelikman et al. (2022) bootstrap reasoning from LLMs, from a few labeled data; Huang et al. (2022) use self-consistency to generate a large number of reasoning traces and fine-tune on them; Zhou et al. (2022) use LLMs themselves to automatically program prompts; Wang et al. (2022); Honovich et al. (2022) use LLMs to generate large instruction datasets for downstream tasks. Collectively, while conceptually related to our work, these previous works deal with a fundamentally different problem, require a more computationally intensive learning procedure (e.g., fine-tuning), or are not fully zero-shot.

**Prompt automation & ICL.**  Numerous methods have been proposed to automate prompt design – USP also endeavors to achieve so by focusing on ICL, a specific component of the prompt. *Soft prompting* methods optimize the embedding space of the LLMs (Li and Liang, 2021; Lester et al., 2021, inter alia) but require gradient access & propagation through massive LLMs and a considerable amount of training data. Recently, various *hard prompting* methods, which search for actual discrete tokens using discrete optimization (Shin et al., 2020; Prasad et al., 2022; Wen et al., 2023), reinforcement learning (Deng et al., 2022; Zhang et al., 2023) and gradient estimation (Diao et al., 2022) have been proposed. While the discrete prompts are more interpretable and (in some cases) compatible with black-box, inference-only LLMs, to our knowledge, none works in the zero-shot setup and tasks beyond CLS problems (with our definition in §3.3) are scarcely investigated. Furthermore, unlike USP, these methods also often require hundreds if not thousands of LLM queries before converging to good prompts. As for ICL, most methods focus on retrieving the best in-context examples from a pool of *golden examples* instead of zero-shot (Rubin et al., 2022; Liu et al., 2022); an exception is AutoCoT which we discuss in §4. Additionally, several other prompting approaches like NPPrompt (Zhao et al., 2022) & Null Prompt (Logan IV et al., 2022) are also proposed, but these

methods again only work for CLS tasks and are orthogonal to USP since they target other aspects of prompting other than the in-context examples.

## B  Datasets and Models

### B.1  Datasets

In this section, we outline the details of the datasets used in this paper.

On PaLM-62B and PaLM-540B, we consider the following datasets. For the CLS tasks, we include commonsense reasoning: boolq (Clark et al., 2019), copa (Roemmele et al., 2011), winogrande (Sakaguchi et al., 2021), ARC easy and challenge (arc_e, arc_c) (Clark et al., 2018), wsc (Levesque et al., 2012); reading comprehension: raceh, racem (Lai et al., 2017); cloze completion: storycloze (Mostafazadeh et al., 2017), natural language inference (NLI): anli-r{1,2,3} (Nie et al., 2020), rte (Wang et al., 2018, 2019), wic (Pilehvar and Camacho-Collados, 2019). For the SFG tasks, we include open-domain QA: web_questions (Berant et al., 2013), natural_questions (Kwiatkowski et al., 2019) and triviaqa_wiki (Joshi et al., 2017); reading comprehension QA: squad (Rajpurkar et al., 2018); word prediction: lambada (Paperno et al., 2016). For the LFG tasks, we include two summarization tasks: xsum (Narayan et al., 2018) and wikilingua (en – *English only*) (Ladhak et al., 2020). Other details of the datasets used in this study are in Table 5.

On PaLM 2 models, we use the BIG-Bench Hard dataset consisting of 23 sub-tasks (data available at https://github.com/suzgunmirac/BIG-Bench-Hard/). The tasks, in alphabetical order, are (the results presented in Fig. 3 and Fig. 9 in App. D are also in the following order):

1. Boolean Expressions
2. Causal Judgment
3. Date Understanding
4. Disambiguation QA
5. Dyck Languages
6. Formal Fallacies Syllogisms Negation
7. Geometric Shapes
8. Hyperbaton (Adjective Ordering)
9. Logical Deduction
10. Movie Recommendation
11. Multi-Step Arithmetic
12. Navigate
13. Object Counting
14. Penguins in a Table
15. Reasoning about Colored Objects
16. Ruin Names
17. Salient Translation Error Detection
18. Snarks
19. Sports Understanding
20. Temporal Sequences
21. Tracking Shuffled Objects
22. Web of Lies
23. Word Sorting

The details of these tasks can be accessed at https://github.com/suzgunmirac/BIG-Bench-Hard/tree/main/bbh. All tasks are converted to SFG format, and the test set of each task consists of 250 test queries. The readers are referred to Suzgun et al. (2022) and the aforementioned GitHub repository for details.

**Licensing**   We outline the license of use of the following datasets:

1. Apache License 2.0: winogrande (https://github.com/allenai/winogrande/blob/master/LICENSE), natural_questions (https://github.com/google-research-datasets/natural-questions/blob/master/LICENSE), triviaqa (https://github.com/mandarjoshi90/triviaqa/blob/master/LICENSE),

2. Academic Free License ("AFL"): piqa (https://yonatanbisk.com/piqa/), BIG Bench datasets (https://github.com/google/BIG-bench/blob/main/LICENSE).

3. MIT: wic, wsc (https://github.com/thoughtbot/superglue/blob/main/LICENSE), xsum (https://github.com/EdinburghNLP/XSum/blob/master/LICENSE), BIG-Bench Hard (https://github.com/suzgunmirac/BIG-Bench-Hard/blob/main/LICENSE)

4. CC0: wikilingua (https://github.com/esdurmus/Wikilingua/blob/master/LICENSE)

5. CC-BY 4.0: storycloze (https://github.com/UKPLab/lsdsem2017-story-cloze/blob/master/LICENSE.txt), rte (https://huggingface.co/datasets/glue), web_questions (https://github.com/brmson/dataset-factoid-webquestions),

| Dataset | Task type (§3.3) | Objective | Test set size $|\mathcal{T}|$ | #classes $|\mathcal{C}|$ | $|\mathcal{D}|/|\mathcal{T}|$ (%) |
|---|---|---|---|---|---|
| winogrande | CLS | commonsense reasoning | 1267 | 2 | 5.05 |
| piqa | CLS | commonsense reasoning | 1838 | 2 | 3.48 |
| storycloze | CLS | commonsense reasoning | 1871 | 2 | 3.42 |
| anlir1 | CLS | NLI | 1000 | 3 | 6.40 |
| anlir2 | CLS | NLI | 1000 | 3 | 6.40 |
| anlir3 | CLS | NLI | 1200 | 3 | 4.53 |
| boolq | CLS | commonsense reasoning | 3270 | 2 | 1.96 |
| copa | CLS | commonsense reasoning | 100 | 2 | 64.0 |
| rte | CLS | NLI | 277 | 2 | 23.1 |
| wic | CLS | context comprehension | 638 | 2 | 10.0 |
| wsc | CLS | commonsense reasoning | 285 | 2 | 22.5 |
| arc_e | CLS | commonsense reasoning | 2365 | 4 | 2.71 |
| arc_c | CLS | commonsense reasoning | 1165 | 4 | 5.49 |
| raceh | CLS | reading comprehension MCQ | 3498 | 4 | 1.83 |
| racem | CLS | reading comprehension MCQ | 1436 | 4 | 4.46 |
| lambada | SFG | word completion cloze | 5153 | n/a | 1.24 |
| web_questions | SFG | open-domain QA | 2032 | n/a | 3.15 |
| natural_questions | SFG | open-domain QA | 3610 | n/a | 1.77 |
| triviaqa_wiki | SFG | open-domain QA | 7993 | n/a | 0.80 |
| squad | SFG | reading comprehension QA | 11873 | n/a | 0.54 |
| xsum | LFG | summarization | 1166 | n/a | 5.49 |
| wikilingua | LFG | summarization | 1500[1] | n/a | 4.27 |

MCQ: multiple choice question. NLI: natural language inference.

[1]Used a random subset of 1500 articles in the validation set.

Table 5: Details of the datasets used in this work for the PaLM models. Note that *test set* here refers to the split of the dataset on which results of this paper are reported – in some datasets, the test labels are not publicly available, and we instead report performance on the dev/validation set. The final column ($|\mathcal{D}|/|\mathcal{T}|$) denotes the percentage of the test set that is used as the unlabelled dataset $\mathcal{D}$ for pseudo-demo generation of USP, AutoCoT and Random demos.

6. CC-BY-SA 3.0: boolq (`https://github.com/google-research-datasets/boolean-questions`,

7. CC-BY-SA 4.0: arc_{c,e} (`https://allenai.org/data/arc`), lambada `https://zenodo.org/record/2630551#.YFJVaWT7S_w`, squad (`https://rajpurkar.github.io/SQuAD-explorer/`),

8. CC-BY-NC 4.0: anli-r{1,2,3} (`https://github.com/facebookresearch/anli/blob/main/LICENSE`)

9. BSD-2: copa (`https://people.ict.usc.edu/~gordon/copa.html`),

10. Unspecified, but allowed "non-commercial research purpose" uses: race_{m,h} (`https://www.cs.cmu.edu/~glai1/data/race/`)

## B.2 Models

We conduct experiments on two PaLM model variants – one with 540 billion parameters (PaLM-540B) and one with 62 billion parameters (PaLM-62B). PaLM is a transformer-based LLM "pre-trained on a high-quality corpus of 780 billion tokens that comprise various natural language tasks and use cases. This dataset includes filtered webpages, books, Wikipedia articles, news articles, source code obtained from open source repositories on GitHub, and social media conversations" (Chowdhery et al., 2022). For the pretraining procedure, PaLM was trained over two TPU v4 Pods with 3072 TPU v4 chips (Chowdhery et al., 2022). In all experiments, we use the quantized PaLM checkpoints (in `int8` precision) for inference only without further pretraining or finetuning.

We also experiment on PaLM 2-M, a variant of the PaLM 2 models (Google et al., 2023). PaLM 2,

a Transformer-based model trained on UL2-like objectives ([Tay et al., 2022](#)), is the successor of PaLM that features stronger multilingual and reasoning abilities.

## C    Implementation Details

### C.1    Prompt Templates

We largely adopt the prompt format used in GPT-3 ([Brown et al., 2020](#)) where possible, and we show the detailed prompt templates in Tables 9, 10 and 11. BBH tasks are formulated as SFG tasks, but they use the CoT prompting templates.

**BBH tasks.**    For experiments using few-shot prompting templates (including few-shot, USP, AutoCoT, and Random demo when the pseudo-demos are acquired), we use following the prompt format to obtain *both* the rationales and the final answers in one prompting step.
*// Demos or pseudo-demos*
```
Q: [QUERY].
A: Let's think step by step. [RATIONALE].
So the answer is [ANS].

...

```
*// Test query*
```
Q: [QUERY].
A: Let's think step by step.
```
For zero-shot experiments (including standard zero-shot, USP, AutoCoT, and Random demo in the stage of acquiring pseudo-demos, we use the following prompt format proposed in [Kojima et al. (2022)](#) to obtain the rationales and answers in two separate steps:
```
Q: [QUERY].
A: Let's think step by step.
```
After the rationales are obtained, the LLM is prompted again to obtain the final answer.
```
Q: [QUERY].
A: Let's think step by step. [RATIONALE].
So the answer is
```

**Non-BBH Tasks.**    It is worth noting that some datasets (raceh, racem and squad) are not zero-shot in their strictest sense even when no demonstration is provided – we follow the GPT-3 prompt format (Fig. G.1, G.3 and G.28 respectively for raceh, racem and squad in [Brown et al. (2020)](#)). In these datasets, each test query consists of a context article and several reading comprehension questions in relation to that article, and even in the

absence of demonstrations (in the form of one or more *other* articles and answered questions associated with those articles), some questions (other than the test question itself) and their solutions to the *same article* are included nevertheless. Therefore, even in the zero-shot setup, the LLM is shown with some demonstration while being "zero-shot" in the sense that the context article itself is novel. Similarly, "$K$ pseudo-demos" in these datasets refer to $K$ (pseudo)-demonstrations, each of which consists of a single article *and their associated questions* (which can be multiple) – in this sense, (1) there are typically more than $K$ *solved questions* prepended to the test queries and (2) even for the model-generated demos, there may be parts of the pseudo-demos that are guaranteed to be correct simply due to the prompting format. Another complication of such a prompt format for methods using pseudo-demos (AutoCoT, Random demo, USP) is that since the responses to a subset of test queries are used as demonstrations themselves, it is possible that *a small number of solutions to some questions are revealed to the LLMs* in the form of solved questions in some demonstrations. However, given that only 5 pseudo-demonstrations are used per question, the impact is insignificant as the test sets of each of these datasets contain thousands to tens of thousands of queries (detailed in Table 5). Furthermore, no method is given *more* unfair advantage over another one, as all methods, including USP and key baselines we compare against, are subject to the same complication, and thus we report results to these datasets nevertheless but mark the impacted results in Tables 2 and 3 with a special note.

### C.2    Additional Experimental Details

**USP.**    USP uses an auxiliary language model for computing the similarity term in Eq. (3). We use Sentence-T5-large ([Ni et al., 2022](#)) for all our experiments. We use a maximum decoding step of 128 tokens for all experiments. For summarization tasks, we apply an additional filtering rule to retain answers whose number of words is between 5 and 90 (to prune out overly short and overly long summaries, which are obviously sub-optimal). For all tasks, we use the following stop tokens as marks for truncation (words after any stop tokens, including the stop tokens themselves, are truncated): "Q:, A:, \n\n" and other special tokens used in PaLM to signal the end of the response. Additionally,

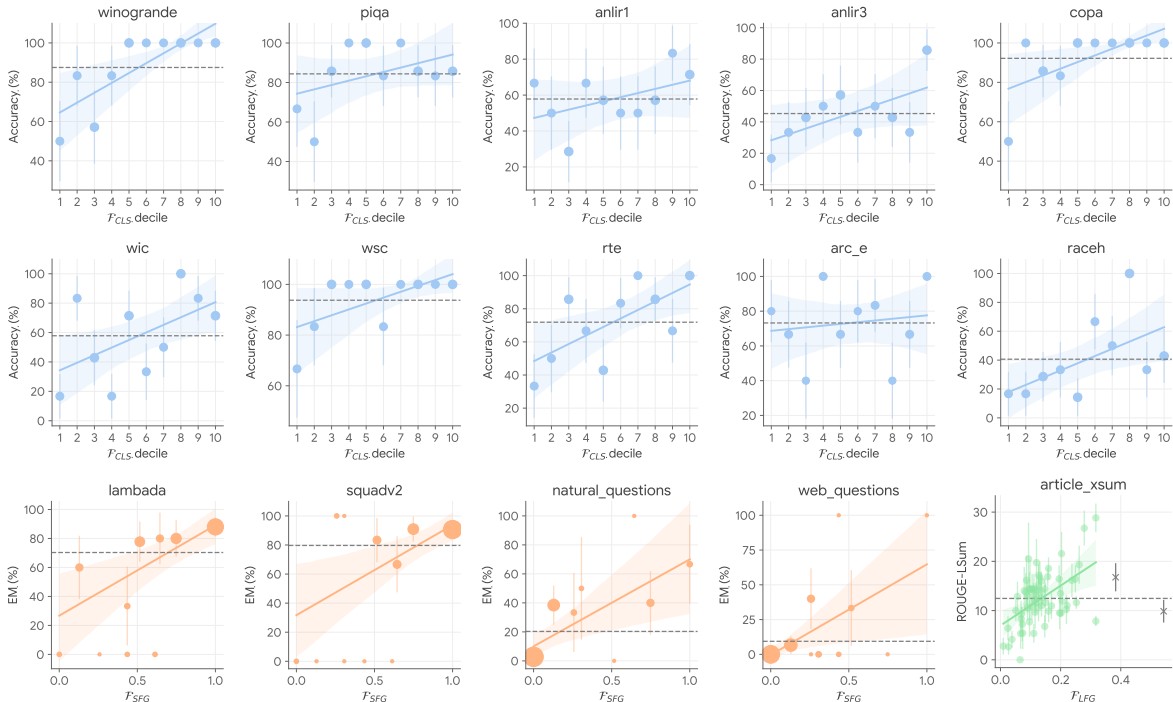

Figure 6: Complementary to Fig. 4, we show the same plot (USP scores vs. ground-truth performance metrics) in additional tasks with PaLM-540B. Refer to Fig. 4 for further explanations.

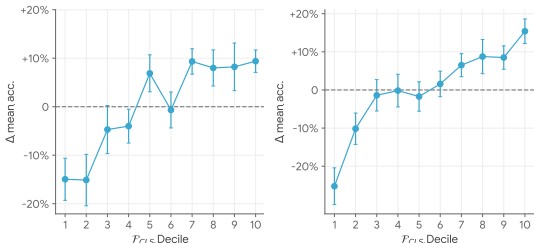

Figure 7: Comparison between the USP score against accuracy averaged across all CLS tasks considered in this paper for PaLM-62B (*left*) and PaLM-540B (*right*). Markers and error bars denote mean ± SEM. It is evident that on expectation, queries with higher USP score tend to be better performing compared to the average model performance (marked by the gray dashed line).

we also apply several additional post-processing steps for the generative tasks, in USP and all other baseline methods:

1. lambada: retain the first output word.

2. squad: remove punctuation marks, remove article words (a, an, the), and retain the portion of the outputs before any newline (\n).

3. web_questions & natural_questions: replace all punctuation marks with a white space, remove article words (a, an, the) and retain the portion of the outputs before any newline (\n)

4. LFG (summarization): since we used the prefix "Article: " at the start of each article to be summarized, we also add "Article: " to the list of stop tokens in addition to the general ones described above.

**Baselines.** We use the same filtering rule for the baseline methods as USP. As discussed, *Random demo* baseline uses an identical procedure to USP, with the sole exception that it does not rely on the scoring functions in 3.3 to select the set of pseudo-demos but rather, *for each test query* $\mathcal{T} = \{x^{(i)}\}_{i=1}^N$, it samples $K$ pseudo-demos randomly from all Stage 1 responses (note that for CLS tasks, it will also follow the procedures described in §3.3 to ensure fair allocation of pseudo-demos across classes). For AutoCoT, we adapt from the official implementation available at https://github.com/amazon-science/auto-cot with a few key modifications: (i) following COSP, we also replace the SentenceBERT (Reimers and Gurevych, 2019) with SentenceT5, a more powerful sentence-level Transformer, for fair comparison against USP; (ii) given that AutoCoT is originally designed for chain-of-thought (CoT) tasks only, we also make necessary modifications such that it is compatible with the general setup. The changes are, in fact, minimal – we only replace the original filtering

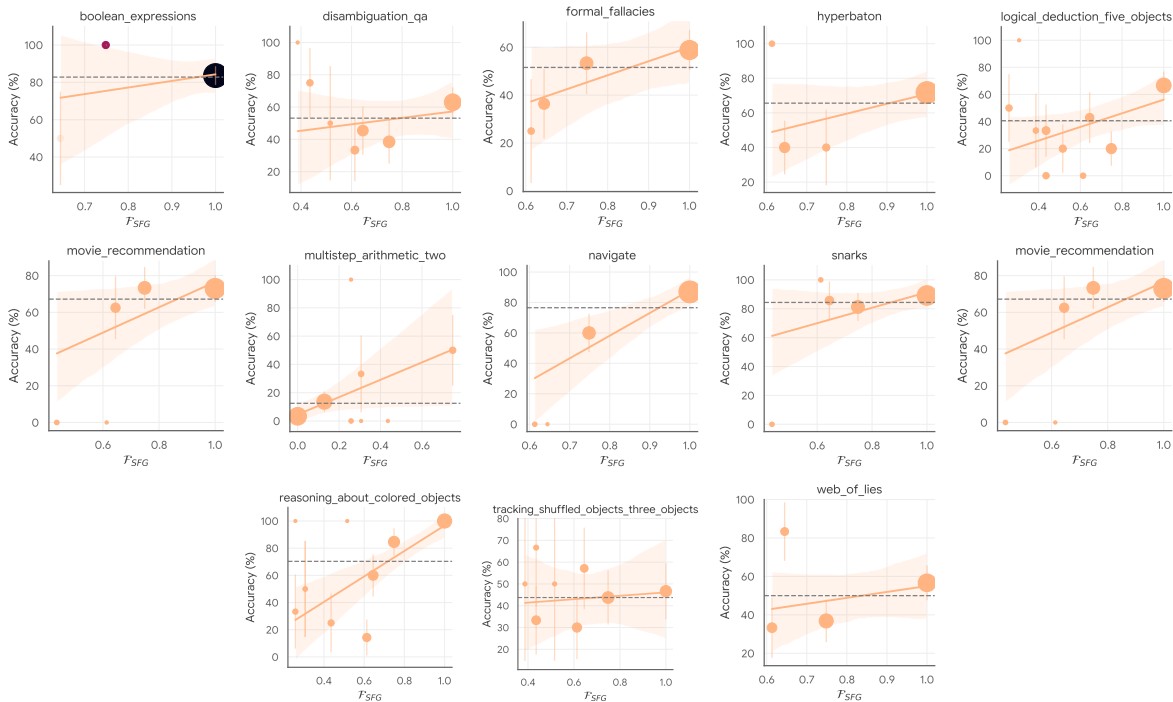

Figure 8: Complementary to Fig. 4, we show the same plot (USP scores vs. ground-truth performance metrics) in additional tasks (BBH tasks with PaLM 2). Refer to Fig. 4 for further explanations.

rules in CoT with the ones we described above for USP. For the few-shot baseline, we closely follow existing works (Brown et al., 2020; Chowdhery et al., 2022) to sample $K$ demonstrations from the training split of each dataset considered, which are prepended to the test queries; we perform sampling for each test query, and thus the choice and order of the demonstrations, in general, differ from one query to another. We use the identical postprocessing rules as USP mentioned in the previous paragraph for the baselines.

## D Additional Experiments

### D.1 PaLM-62B Results

In this section, we show the PaLM-62B results in Tables 6, 7 and 8.

### D.2 Few-shot USP

In this section, we show the results of applying USP in the few-shot setup. We conduct experiments on the BBH datasets with the PaLM 2-M model, as in the main text. Instead of using zero labeled samples (0-shot in Table 3) or 3 labeled samples (3-shot, or few-shot in Fig. 3), we use 1 labeled sample per query, and use USP to generate 2 further pseudo-demos (we name this variant **USPfs** where *fs* stands for "few-shot") – this is to emulate the setup where scarce labeled data are available and

| Setting | Zero-shot | | | | Few-shot |
|---|---|---|---|---|---|
| Method | 0-shot | Auto-CoT | Random demo | USP *(Ours)* | |
| winogrande | 76.95 | **80.19** | **80.19** | **80.98** | 77.35 |
| piqa | 79.87 | 80.58 | **80.85** | 80.74 | **81.07** |
| storycloze | 80.28 | 82.84 | 82.68 | **85.03** | 84.23 |
| anlir1 | 37.20 | 36.80 | **40.70** | **41.90** | 39.30 |
| anlir2 | 38.10 | 38.10 | **39.20** | 37.00 | **38.20** |
| anlir3 | 37.17 | 39.58 | 42.58 | **45.75** | 40.17 |
| boolq | 84.86 | 82.84 | 85.44 | **85.90** | 83.82 |
| copa | **94.00** | 92.00 | **93.00** | 92.00 | 91.00 |
| rte | 67.87 | **79.42** | 76.53 | **76.53** | **76.53** |
| wic | 49.53 | 55.33 | 49.53 | 49.53 | **58.13** |
| wsc | 86.67 | 87.02 | 87.02 | **89.82** | 83.51 |
| arc_e | 76.58 | **81.61** | 79.62 | **82.49** | 80.72 |
| arc_c | 48.24 | **51.07** | 49.61 | 46.95 | **51.16** |
| raceh* | 44.77 | **46.51** | 44.65 | 45.60 | 45.54 |
| racem* | 60.65 | **64.42** | 63.44 | 64.48 | 63.30 |
| *Average ↑* | *64.18* | *66.55* | *66.34* | ***66.98*** | *66.31* |
| *Gain over 0-shot (%)* | *0.00* | *3.70* | *3.36* | ***4.36*** | *3.31* |
| *Average rank ↓* | *4.07* | *2.73* | *2.60* | ***2.20*** | *2.87* |

Table 6: **Accuracy** on **CLS tasks** (Table 1) with PaLM-62B. *See notes in App. C.1.

it is desirable to use USP to augment the set of golden demonstrations.

We show the results in Fig. 9: we find that while using 3 golden examples is still the best, USPfs outperforms both standard 1-shot and USP without using any labeled example, and it also closes roughly half of the gap between 1-shot and 3-shot – this suggests that USP routine continues to be effective in few-shot setup, and thus can also be suitable for the setups less strict than zero-shot, but where obtaining many human-labeled demonstrations is still expensive or otherwise challenging.

| Model | Setting | Zero-shot | | | | Few-shot |
|---|---|---|---|---|---|---|
| | Method | 0-shot | Auto-CoT | Random demo | USP *(Ours)* | 5-shot |
| PaLM 62B | lambda[a] | **75.61** / - | 73.74 / - | 73.57 / - | **74.38** / - | 74.17 / - |
| | web_questions | 12.30 / 25.98 | 18.21 / 36.33 | 17.96 / 33.65 | **20.37 / 36.62** | **27.76 / 42.90** |
| | natural_questions | 18.45 / 27.29 | 21.60 / 30.80 | 20.39 / 29.90 | **23.85 / 33.69** | **27.59 / 37.39** |
| | triviaqa_wiki | 67.71 / 72.85 | 69.49 / 74.17 | **70.43 / 74.84** | 69.84 / 74.14 | 62.11 / 67.29 |
| | squad* | 69.59 / 75.34 | **85.11 / 89.14** | 80.30 / 84.88 | 83.63 / 87.88 | 79.85 / 83.96 |
| | *Average ↑* | *48.73 / 55.41[b]* | *53.63 / 60.84[b]* | *52.53 / 59.37[b]* | *54.41 / 61.34[b]* | *54.30 / 61.14[b]* |
| | *Gain over 0-shot (%)* | *0.00 / 0.00[b]* | *10.05 / 9.79[b]* | *7.79 / 7.13[b]* | *11.66 / 10.70[b]* | *11.42 / 10.34[b]* |
| | *Average rank ↓[c]* | *4.00* | *2.80* | *3.40* | *2.00* | *2.80* |

Table 7: **Exact Match (EM) / F1** on SFG tasks with PaLM-62B [a]Only EM shown as lambda expects a single correct answer. [b]Used lambda EM for the average F1 score. [c]Ranked in terms of EM. *See notes in App. C.1. Refer to Table 2 for further explanations.

| Model | Setting | Zero-shot | | | | Few-shot |
|---|---|---|---|---|---|---|
| | Method | 0-shot | Auto-CoT | Random demo | USP *(Ours)* | 1-shot |
| PaLM 62B | xsum | 17.7 / 14.1 / 0.183 | 19.8 / 15.5 / 0.338 | 19.1 / 15.3 / 0.317 | 21.9 / 17.1 / **0.347** | **24.3 / 19.1 / 0.337** |
| | wikilingua (en) | 20.1 / 16.3 / 0.416 | 10.6 / 9.0 / 0.333 | 18.3 / 14.6 / 0.396 | **28.6 / 23.3 / 0.486** | 27.5 / 22.0 / **0.488** |
| | *Average ↑* | *18.9 / 15.2 / 0.299* | *15.2 / 12.3 / 0.336* | *18.7 / 14.9 / 0.357* | *25.3 / 20.2 / 0.417* | *25.9 / 20.5 / 0.413* |
| | *Gain over 0-shot (%)* | *0.0 / 0.0 / 0.0* | *-19.5 / -19.1 / 12.0* | *-1.0 / -1.5 / 19.1* | *34.0 / 33.0 / 39.1* | *37.4 / 35.3 / 37.7* |

Table 8: **ROUGE-1 / ROUGE-Lsum / BLEURT (Sellam et al., 2020) scores** on LFG tasks with PaLM-62B. Refer to Table 2 for further explanations.

## D.3 Additional Comparison Between USP Scores and Ground-truth Quality

Complementary to Fig. 4 in §5, we show plots of the same relation for other tasks considered in PaLM-540B in Fig. 6, and the aggregated results (across CLS tasks) in Fig. 7; we also show the comparisons on selected BBH tasks with PaLM 2-M in Fig. 8 These give further evidence that USP heuristic described in §3.3 selects higher quality demonstrations in comparison to the average model performance.

## D.4 Examples of Selected Pseudo-demos

We show some examples of the pseudo-demos generated by USP on a variety of representative tasks in Table 12.

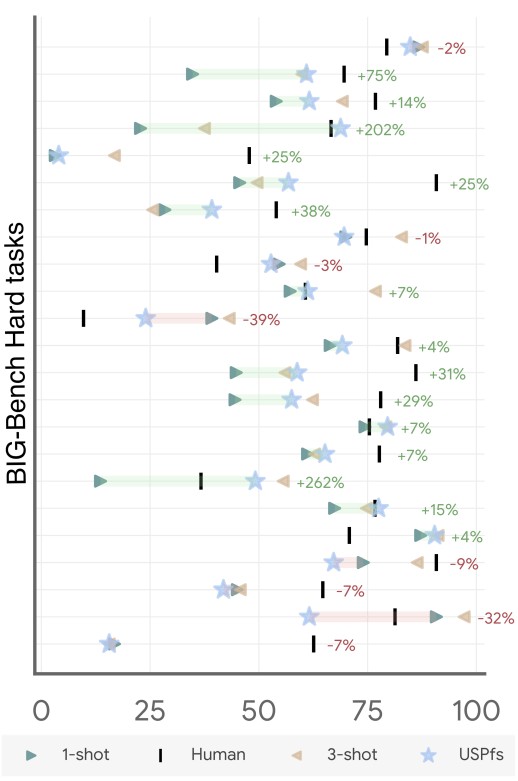

| Setting | Zero-shot | Few-shot | | |
|---|---|---|---|---|
| Method | USP (Ours) | 1-shot | USPfs (Ours) | 3-shot |
| Average ↑ | 54.18 | 51.20 | **55.80** | **60.36** |
| Gain over 1-shot (%) | 5.82 | 0.00 | **9.00** | **17.89** |
| Average rank ↓ | 2.61 | 3.06 | **2.44** | **1.89** |

Figure 9: Few-shot **accuracy** on BIG-Bench Hard tasks with PaLM 2-M (each line represents a task – refer to App. B for full details). The gain/loss of USP over standard **1-shot** is shown in percentages. USPfs generates 2 pseudo-demos on top of the 1 provided golden demo. Standard zero-shot USP and 3-shot results are reproduced from Fig. 3.

| Dataset | Prompt template |
|---|---|
| winogrande | The woman avoided the hole but easily stepped over the pit over the {hole / pit}, because the hole was very shallow |
| piqa | Q: To pour hot fudge over ice cream before serving,\nA: { pour the hot fudge over ice cream that has just been pulled from the freezer and scooped out of it's container with an ice cream scoop into a bowl / pour the hot fudge over ice cream that has been pulled out of the freezer and softened for fifteen minutes, then scooped out of it's container with an ice cream scoop into a bowl. } |
| storycloze | Neil wanted to see ancient temples and ruins. He decided Asia was a great place to start. He flew to Cambodia and went sightseeing. He saw so many old temples in the jungles there. {Neil was bored of the trip and went home. / Neil was happy he made the trip.} |
| anlir{1,2,3} | Lofar is a Telugu film directed by Puri Jagannadh. It features Varun Tej and Disha Patani in the lead roles while Revathi and Posani Krishna Murali appear in crucial supporting roles. The film was officially launched on 8 July 2015 in Hyderabad. Earlier makers revealed the first look posters and trailer of the movie which received good response in the social media.\nquestion: Varun Tej had billing over Disha Patani in Lofar. Is it true, false, or neither?\nanswer: {true / false / neither} |
| boolq | Evil Queen (Disney) – This version of the fairy tale character has been very well received by film critics and the public, and is considered one of Disney's most iconic and menacing villains. Besides in the film, the Evil Queen has made numerous appearances in Disney attractions and productions, including not only these directly related to the tale of Snow White, such as Fantasmic!, The Kingdom Keepers and Kingdom Hearts Birth by Sleep, sometimes appearing in them alongside Maleficent from Sleeping Beauty. The film's version of the Queen has also become a popular archetype that influenced a number of artists and non-Disney works.\nquestion: are maleficent and the evil queen the same\nanswer: {yes / no} |
| copa | The tree branch landed in the river {so the branch moved downstream. / the river's current became stronger.} |
| rte | Tropical Storm Irene on August 11, 2005 at 16:15 UTC. Tropical Storm Irene will increase in strength over the next several days, possibly developing into a hurricane that will hit the east coast of the United States, said the National Hurricane Center of Miami, Florida in a report today. Irene was located approximately 975 kilometers south-southeast of Bermuda at 16:00 UTC today. Forecasters say that the storm is now moving in a west- northwest direction with top sustained winds of 40 miles per hour.\nquestion: A storm called Irene is going to approach the east coast of the US. Is it true or false?\nanswer: {true / false} |
| wic | Had unusual longevity in the company.\nHer longevity as a star.\nquestion: is the word 'longevity' used in the same way in the two sentences above?\nanswer: {Yes / No} |
| wsc | {The city councilmen refused the demonstrators a permit because The demonstrators / The city councilmen refused the demonstrators a permit because The city councilmen} feared violence. |
| arc_{c,e} | Q: Which tool should be used to measure the stem length of a plant?\nA: {a balance / a metric ruler / a graduated cylinder / a thermometer} |
| race{h,m} | 'Article: October is getting closer and it also means that the year of 2014 is coming to an end. "Hooray! It's a holiday!" While you are thinking of putting textbooks aside and playing video games, let's take a look at what children in other continents usually do during their holidays. Children in America don't have much homework to do. They keep themselves busy by playing camp games. A parent says, "My daughter Shirley usually attends different camps. We don't ask her to spend plenty of time on maths problems or spelling tests." Children in Australia take partin activities on over twenty different themes . They learn painting, dancing, singing, history, culture and so on. Parents can _ their kids to enjoy the learning process and to build a closer relationship with them. These are what African kids do: build a boat, have a camel race, make a drum and make a rag football. Don't you think it is interesting that kids in other places have no idea how to make a drum, but kids in Africa do? Plan your holiday well and try what you want to try. Make a good plan and you will have a lot of fun. Q: Where does Shirley come from? A: {America, China, Brazil, Australia} |

Table 9: Prompt templates (with examples) of the CLS datasets used. Note that the anlir{1,2,3}, race{m,h}, arc_c,e datasets are grouped together due to similar prompt format. The LLM is asked to output the log-likelihood using each of the options marked in blue as a possible text completion, and the option with the highest predicted probability is selected as the final prediction. Note that the race_{h,m} datasets are not strictly zero-shot as the prompt already contains several answered questions to the context passage leading up to the text query – see App. C.1 for detailed explanations.

| Dataset | Prompt template |
|---|---|
| lambda | `Yes, I am absolutely sure you did, Cook. I can see the empty egg boxes like you said, thirteen of them."\nCaptain Porter was used to getting to the bottom of these sorts of incidents, especially when it involved some of his boys.\n"Has anyone else been in the kitchen,` `Cook` |
| web_questions | `Q: who were jesus siblings?\nA:`{Jude the Apostle / James the Just / Simon (brother of Jesus) / Joses} |
| natural_questions | `Q: how long is the bridge between new brunswick and prince edward island\nA:` `2.9-kilometre` |
| triviaqa_wiki | `Q: How many medals did the United States win at the 2010 Winter Olympics?\nA:`{37 / thirty seven} |
| squad | `Title: Southern_California\n\nBackground: The San Bernardino-Riverside area maintains the business districts of Downtown San Bernardino, Hospitality Business/Financial Centre, University Town which are in San Bernardino and Downtown Riverside.\n\nQ: The Sand Bernardino - Riverside area maintains what kind of district?\n\nA: business\n\nQ: Other than San Bernardino, what is the name of the other city that maintains the districts including University Town?\n\nA: Riverside\n\nQ: Other than Downtown San Bernardino, and University Town, what is the name of another business district in the San Bernardino-Riverside area?\n\nA: Hospitality Business/Financial Centre\n\nQ: What business districts does the San Bernardino area maintain?\n\nA: no answer\n\nQ: What business districts does the Riverside area maintain?\n\nA:` `no answer` |

Table 10: Prompt templates (with examples) of the SFG datasets used. The expected response(s) are marked in green. Note that the squad dataset is not strictly zero-shot as the prompt already contains several answered questions to the context passage leading up to the text query – see App. C.1 for detailed explanations.

| Dataset | Prompt template |
|---|---|
| xsum | `Article: Upsetting events often make the news because they don't happen very often. \nThis section gives you some tips about what to do if you are feeling sad about what you've seen, heard or read.\nYou can rely on Newsround to tell you the important facts about a story - but some things you hear might be a bit scary or make you feel worried.\nRemember that worrying stories are often in the news because they are rare - they don't happen very often.\nIt is incredibly unlikely that what you're reading about or watching might happen near you.\nDiscuss the stories with your parents or friends. You'll feel better that you're not the only one worried. \nYou could also talk to your teacher about it - maybe you could have a class discussion which would help you understand the issue better.\nIf you're having nightmares or trouble sleeping because of something you've heard in the news: \n\ntl;dr:` `Some stories reported by Newsround can make you feel sad - but you are not the only one and it's OK to have those feelings.` |
| wikilingua | `Article: The most commonly used classes of OTC pain medications include Acetaminophen (Tylenol), and a class of drugs called "NSAIDs." NSAIDs stand for "nonsteroidal anti-inflammatory drugs," and include medications such as Ibuprofen (Advil, Motrin), and Naproxen sodium (Aleve). Aspirin is also technically an NSAID, although it more frequently used in the prevention of heart attacks and strokes than it is in easing chronic pain.` `[Omitted]` `This can lead to gastrointestinal bleeding and anemia. Special care should be taken with those who drink alcohol. Always read the label of cold and flu medications carefully to see what ingredients are present in the mixture. If you need OTC drugs for more than 10 days, book an appointment with your physician to do a more detailed assessment of your pain, and to look into alternative modes of treatment that may be more effective (and also safer) for you moving forward. Also consult your doctor if you have other health concerns, such as ongoing heart disease, kidney disease, or liver disease, prior to using OTC medications for your pain.\n\ntl;dr:` `Be aware of acceptable doses of OTC pain medications. Understand the risks of overusing OTC drugs. Consult your doctor if you are unable to manage your pain without exceeding the recommended daily dosage of OTC drugs.` |

Table 11: Prompt templates (with examples) of the LFG datasets used. The reference summaries are marked in orange. We tried various prompts to elicit zero-shot summarization ability in PaLM and found that "\n\ntl;dr:" works the best, likely because it is a common shorthand used online forums where most of the PaLM pretraining data were obtained.

| Dataset | Generated pseudo-demos |
|---|---|
| storycloze | 'I love my job more than anything in the world. I work from home as a freelance artist. I work in my sweatpants and get to draw and paint. People pay a lot of money for my art and I am high in demand. ==I enjoy it so much, it doesn't even feel like work==.

My friend got me a planter as a housewarming gift. It's small and cute and fits in nicely with my decor. But I'm still not sure where to put it. I hope he doesn't get sad. ==I'm sure I'll figure out a place for it eventually==.' |
| anlir3 | 'Chinese
Sally bought a book from the library. She opened it to page 3. She read the words but they didn't make since to her. She looked at the cover. She got a Chinese book by accident. question: Sally was able to read Chinese. Is it true, false, or neither? answer: ==neither==.

TORONTO, March 7 (Reuters) - The Canadian dollar weakened to a session low against the greenback after data showed the domestic economy unexpectedly shed jobs in February. At the same time, investors were also taking in data south of the border that showed U.S. job growth accelerated last month. The Canadian dollar was at C\$1.1055 to the greenback, or 90.46 U.S. cents, weaker than Thursday's close of C\$1.0992, or 90.98 U.S. cents. The loonie hit a session low of C\$1.1064 shortly after the data was released. question: Toronto is the most populous city in Canada. Is it true, false, or neither? answer: ==true==.

A Girl Name Reagan
Tim was asked to show the new student around. He was to wait in the school office for a student named Reagan. Waiting for the student to arrive he wondered what they would be like. Tim assumed the person would be tall like him and a boy. When the person finally arrived it was short girl dressed all in blue. question: Tim assumed the person would be a girl. Is it true, false, or neither? answer: ==false==.' |
| natural_questions | 'Q: when was rosencrantz and guildenstern are dead written A: ==1966==.

Q: where was the statue of liberty originally built A: ==france==.' |
| triviaqa_wiki | Q: In the 2005 remake of the film 'King Kong' who played the part of Ann Darrow, originally played by Fay Wray? ==A: naomi watts.==

Q: In which contact sport do two rikishi compete inside a dohyo ? A: ==sumo==.' |
| wikilingua | 'Article: In order to scan a QR code with your iPhone or iPad camera, you must first update your iPhone or iPad to iOS 11 or later. Then open Settings . Tap the grey app with gears on it. You'll typically find this app on the Home Screen. Scroll down and tap Camera. This option is about halfway down the Settings page. Tap the white "Scan QR Codes" switch. It will turn green. Doing so will enabled your iPhone's or iPad's camera's QR code scanner. If the "Scan QR Codes" switch is already green, your iPhone or iPad is ready to scan QR codes. Tap the Camera app icon, which resembles a black camera on a grey background. You can also swipe up from the bottom of the screen to open the Control Center and then tap the camera icon there. The QR code should be centered in the middle of the iPhone or iPad screen, with no edges or pieces off-screen. If your camera opens to the front-facing camera, first tap the camera with arrows icon in the bottom-right corner of the screen. Once it does, a grey notification that says something like "Open [website] in Safari" will appear at the top of the screen. If the code contains a website URL, doing so will open the QR code's website in your iPhone's or iPad's Safari browser. tl;dr: ==1. Update to iOS 11 or later. 2. Open Settings. 3. Tap Camera. 4. Tap the white "Scan QR Codes" switch. 5. Open the Camera app. 6. Center the QR code in the camera's view. 7. Tap the notification that appears at the top of the screen.==' |

Table 12: Examples of generated pseudo-demos from USP on representative tasks (PaLM-540B). The response parts of the pseudo-demos are highlighted: ==correct answers==; wrong answers; In LFG problems, there is no single, correct answer. We instead simply highlight the solution in ==yellow==.

| Dataset | Generated pseudo-demos |
|---|---|
| boolean_expressions | Q: False and ( False or not False ) is
A: Let's think step by step. not False is True. So the expression is False and ( False or True ). Since False or True is True, the final answer is False. So the answer is False. |
| object_counting | Q: I have four carrots, a cabbage, an onion, a head of broccoli, a yam, a stalk of celery, a lettuce head, a potato, and three cauliflowers. How many vegetables do I have?
A: Let's think step by step. 4 + 1 + 1 + 1 + 1 + 1 + 1 + 1 + 3 = 15.
So the answer is 15
*Authors' note: an example where a final arithmetic error results from a correct reasoning trace.* |
| tracking_shuffled_objects | Q: Alice, Bob, Claire, Dave, and Eve are on the same team in a soccer match. At the start of the match, they are each assigned to a position: Alice is playing left midfielder, Bob is playing goalkeeper, Claire is playing fullback, Dave is playing striker, and Eve is playing left winger.
As the game progresses, pairs of players occasionally swap positions. First, Eve and Dave trade positions. Then, Dave and Alice trade positions. Then, Bob and Dave trade positions. Then, Claire and Bob trade positions. Finally, Claire and Alice trade positions. At the end of the match, Bob is playing
Options:
(A) left midfielder
(B) goalkeeper
(C) fullback
(D) striker
(E) left winger
A: Let's think step by step. After Eve and Dave trade positions, Eve is playing striker and Dave is playing left winger. After Dave and Alice trade positions, Dave is playing left midfielder and Alice is playing striker. After Bob and Dave trade positions, Bob is playing striker and Dave is playing fullback. After Claire and Bob trade positions, Claire is playing striker and Bob is playing fullback. After Claire and Alice trade positions, Claire is playing left midfielder and Alice is playing striker. Therefore, at the end of the match, Bob is playing fullback.
So the answer is (C). |
| ruin_names | Q: Which of the following is a humorous edit of this artist or movie name: 'the girl with all the gifts'?
Options:
(A) the girl with all the grifts
(B) the girl with alf the gifts
(C) the girl with iall the gifts
(D) the girl with all thke gifts
A: Let's think step by step. The girl with all the grifts means the girl with all the cons. Grifts means cons. Grifts is a funny word. Thus, the answer is (A). |
| disambiguation_qa | Q: In the following sentences, explain the antecedent of the pronoun (which thing the pronoun refers to), or state that it is ambiguous.
Sentence: The pathologist obtained tissue samples from the victim to look at under their microscope.
Options:
(A) It was the pathologist's microscope
(B) It was the victim's microscope
(C) Ambiguous
A: Let's think step by step. Ambiguous because it is unclear if the pathologist or the victim owns the microscope.
So the answer is (C)
*Authors' note: the ground-truth is (A), but we believe that the LLM's rationales & final solutions are also reasonable.* |

| | |
|---|---|
| movie_review | Q: Find a movie similar to Star Wars Episode V - The Empire Strikes Back, The Lord of the Rings The Fellowship of the Ring, American Beauty, Forrest Gump:
Options:
(A) Upside Down The Creation Records Story
(B) The Adventures of Sherlock Holmes and Doctor Watson
(C) Waking Life
(D) The Lord of the Rings The Two Towers
A: Let's think step by step. The Lord of the Rings The Two Towers is the second part of The Lord of the Rings film trilogy. The Lord of the Rings The Fellowship of the Ring is the first part of The Lord of the Rings film trilogy. The Lord of the Rings The Two Towers is similar to Star Wars Episode V - The Empire Strikes Back. The answer: (D) |
| snarks | Q: Which statement is sarcastic?
Options:
(A) People never foment wars to relinquish territory
(B) People never foment wars to conquer territory
A: Let's think step by step. Wars are usually fought to conquer territory. The answer is (A) |

Table 13: Examples of generated pseudo-demos from USP on representative BBH tasks (PaLM 2-M). The response parts of the pseudo-demos are highlighted: correct answers; wrong answers; partially correct answers, or questionable/irrelevant reasoning in the generated rationales that are not wrong