# OpenReview forum: "Universal Self-Adaptive Prompting"
_EMNLP/2023/Conference — EMNLP 2023 Main_

### Official Review · Reviewer_uuzn · 2023-08-02

**Soundness:** 4

**Excitement:**

4: Strong: This paper deepens the understanding of some phenomenon or lowers the barriers to an existing research direction.

**Missing References:**

L40 “Indeed, one of the most prominent and fundamental emerging abilities of modern LLMs is their impressive zero-shot generalizability”. Actually, it is debatable how really emergent these abilities are instead of being a by-product of the training curricula. This was the original research question for instruction-tuning papers like T0 and FLAN. I suggest rephrasing the sentence or citing something else to support the statement.

**Paper Topic And Main Contributions:**

The paper tackles the problem of zero-shot learning in language models (LMs) by proposing a procedure to automatically design prompts and increase performance on downstream tasks. Results show that the method is effective on several different types of tasks compared to previous approaches.

**Reasons To Accept:**

The paper is well-written, the method is clear and the experiments are sound

**Reasons To Reject:**

The approach has been tested only with PALM and PALM2 which are two proprietary LMs not particularly accessible (low reproducibility). Furthermore, although it is reasonable to think that it will work with other types of models, there is no experiment in this direction.

**Reproducibility:**

2: Would be hard pressed to reproduce the results. The contribution depends on data that are simply not available outside the author's institution or consortium; not enough details are provided.

**Reviewer Confidence:**

4: Quite sure. I tried to check the important points carefully. It's unlikely, though conceivable, that I missed something that should affect my ratings.

**Typos Grammar Style And Presentation Improvements:**

Figure 3 is not particularly readable. Labels are truncated and there is too much information on the plot, making the figure difficult to parse.

---

> ### Author Rebuttal · Authors · 2023-08-26
>
> We thank the reviewer for their positive and constructive feedback! Please see our detailed response below.
>
> > The approach has been tested only with PALM and PALM2 which are two proprietary LMs not particularly accessible (low reproducibility). Furthermore, although it is reasonable to think that it will work with other types of models, there is no experiment in this direction.
>
> We acknowledge the reviewer’s concerns. We note that 1) while the LLMs are proprietary, the technical details of PaLM and PaLM 2 are both publicly available (PaLM 2 also has a [public API](https://developers.generativeai.google/), and they are both based on similar architectures that open-sourced LLMs are built on, and thus, as the reviewer mentioned, there is no reason why USP won’t work with other types of models and 2) we will try to include experiments in some of the more accessible models like Llama to alleviate the reviewer’s concern on reproducibility.
>
> > L40 “Indeed, one of the most prominent and fundamental emerging abilities of modern LLMs is their impressive zero-shot generalizability”. Actually, it is debatable how really emergent these abilities are instead of being a by-product of the training curricula. This was the original research question for instruction-tuning papers like T0 and FLAN. I suggest rephrasing the sentence or citing something else to support the statement.
>
> We understand the reviewer’s concerns – when we said “impressive zero-shot generalizability,” we were referring to the fact that modern LLMs tend to be more generally capable of following natural language instructions better in solving tasks they have not explicitly seen before. As the reviewer aptly pointed out, it is indeed debatable and an open question on what exactly causes it. We will rephrase this statement to make our intention clearer and less controversial, as the reviewer suggested.
>
> > Figure 3 is not particularly readable. Labels are truncated and there is too much information on the plot, making the figure difficult to parse.
>
> We thank the reviewer for their suggestion – we will amend the figure to make it more readable and move some of the redundant information to the supplementary materials.

---

### Official Review · Reviewer_GeRr · 2023-08-05

**Soundness:** 3

**Excitement:**

4: Strong: This paper deepens the understanding of some phenomenon or lowers the barriers to an existing research direction.

**Paper Topic And Main Contributions:**

The paper's main contribution is the introduction of Universal Self-Adaptive Prompting (USP), a novel automatic prompt design approach specifically designed for zero-shot learning, but also compatible with few-shot scenarios. The USP method requires only a small amount of unlabeled data and an inference-only LLM, making it highly versatile.

**Questions For The Authors:**

Question 1A: Could you provide more clarity on the Universal Self-Adaptive Prompting (USP) method's implementation details? For example, what computational resources are required, and how does this compare to existing methods?

Question 1B: Could you provide a more detailed comparison of the USP method with other state-of-the-art methods? It would be helpful to understand how USP performs in comparison, both in terms of accuracy and computational efficiency.

Question 1C: The paper mentions that the USP method requires only a small amount of unlabeled data. Could you provide more specifics on the amount of data used in your experiments and how this impacts the performance of the model?

Question 1D: Are there any limitations or potential drawbacks to the USP method that were not discussed in the paper? If so, could you elaborate on these and how they might be addressed in future work?

Question 1E: Could you elaborate on the scalability of the USP method? Specifically, how does the performance of the method change with the size of the language model and the amount of available data?

Question 1F: Are there any specific types of tasks or domains where the USP method performs particularly well or poorly? If so, could you provide some examples and explain why this might be the case?

**Reasons To Accept:**

Novel Approach: The paper presents a new approach called Universal Self-Adaptive Prompting (USP), which is designed to improve the zero-shot learning capabilities of large language models (LLMs). This is a significant contribution to the field of NLP.
Versatility: The proposed USP method requires only a small amount of unlabeled data and an inference-only LLM. This shows that the method is versatile and potentially widely applicable.
Comprehensive Evaluation: The authors have evaluated the USP method using PaLM and PaLM 2 models across more than 40 different tasks. The results indicate that the USP method significantly improves performance over standard zero-shot baselines and is often comparable to or even superior to few-shot baselines.
Clarity: The paper is well-structured and clearly written, which makes it easier to understand the proposed method and its implications.

**Reasons To Reject:**

Limited Explanation of Methodology: Although the paper presents a new method, it might not provide a comprehensive explanation of the methodology or a detailed comparison with existing methods. This could make it challenging for readers to fully understand the novelty and value of the proposed approach.
Unclear Implementation Details: The paper might lack a clear explanation of the practical aspects of implementing the USP method, such as the computational resources required.
Lack of Comparative Analysis: While the paper shows that the USP method outperforms standard zero-shot baselines, it might not provide a detailed comparison with other state-of-the-art methods. This could make it difficult to assess the relative performance of the USP method.

**Reproducibility:**

4: Could mostly reproduce the results, but there may be some variation because of sample variance or minor variations in their interpretation of the protocol or method.

**Reviewer Confidence:**

3: Pretty sure, but there's a chance I missed something. Although I have a good feel for this area in general, I did not carefully check the paper's details, e.g., the math, experimental design, or novelty.

**Typos Grammar Style And Presentation Improvements:**

Clarity of Methodology: The explanation of the Universal Self-Adaptive Prompting (USP) methodology could be clearer. More details and perhaps a step-by-step walkthrough of the approach would be helpful.

Comparisons with Other Methods: The paper could benefit from a more detailed comparison with existing methods. This could include a table summarizing the main differences and similarities, as well as a discussion of the advantages and limitations of each method.

Use of Figures: If not already included, the use of illustrative figures to explain the USP method could improve the readability of the paper and help readers understand the approach better.

Data Details: More information on the datasets used in the study would be helpful. This could include the size of the datasets, how they were collected, and any preprocessing steps.

Discussion of Results: The paper could provide a more detailed discussion of the results, including potential reasons for the observed performance and any surprising or unexpected findings.

Conclusions and Future Work: The conclusion section could provide a more comprehensive summary of the main findings, their implications, and potential directions for future research.

---

> ### Author Rebuttal · Authors · 2023-08-26
>
> We thank the reviewer for their detailed feedback and respond to their concerns and questions below. We believe that we have thoroughly addressed their concerns, and would be grateful if they could consider increasing their rating if they found our response satisfactory.
>
> > Limited Explanation of Methodology: Although the paper presents a new method, it might not provide a comprehensive explanation of the methodology or a detailed comparison with existing methods. This could make it challenging for readers to fully understand the novelty and value of the proposed approach.
>
> > Unclear Implementation Details: The paper might lack a clear explanation of the practical aspects of implementing the USP method, such as the computational resources required.
>
> > Question 1A: Could you provide more clarity on the Universal Self-Adaptive Prompting (USP) method's implementation details? For example, what computational resources are required, and how does this compare to existing methods?
>
> **Explanation of methodology and implementation details.**
> We have discussed all inputs and requirements of USP in Algorithm 1, Page 4. Namely, one needs a test set, an unlabeled set (which can be a subset of the test set, the test set itself, or another related set of unlabeled queries), the task type (CLS, SFG, or LFG), and the LLM itself (given the LLM outputs, computing USP scores is trivial on any modern PC without GPUs). *All additional implementation details are included in Appendix B and C*. We would be grateful if the reviewer could specifically point out what, in their opinion, is not clear.
>
> **Computational costs.**
> Computing the USP scores itself is cheap (can be done in any modern PC/laptop with CPU only), and the cost is thus bottlenecked by the amount of processing from the LLM side. In particular, the additional costs are:
>
> - Stage 1: Since we use 64 unlabeled samples, we only require 64 additional model queries for the CLS task and $64m$ (we use $m = 6$ in paper) for SFG and LFG tasks – note that we can also use batching to parallelize this step. The additional cost can be captured by the $|\mathcal{D}|\/|\mathcal{T}|$ (the fraction of the unlabeled samples to the size of the entire test set) column in Table 5 in the Appendix, which as we can see, the cost is almost always negligible compared to the cost we need to incur anyway (except for some very small-scale toy tasks with small test tasks)
> - Stage 2: This stage is completely identical to standard few-shot in-context learning.
>
> **Cost comparison to existing methods**
> - *Zero-shot*: USP requires the additional Stage 1, which typically only adds a small amount of cost, as discussed above. In Stage 2, the LLM needs to process a longer context due to the use of pseudo-demos for in-context learning. However, this is due to the use of in-context learning (ICL) and is not an additional cost uniquely attributable to USP – it is true for all other methods (standard few-shot, AutoCoT, and Random Demo) relying on ICL.
>
> - *Few-shot*: The only additional overhead is the use of Stage 1, but crucially, no labeled data is required at any point in time.
>
> - *AutoCoT* [1]: USP features an identical cost to AutoCoT in Stage 2. The difference is that USP is slightly more expensive in Stage 1 as USP performs LLM forward passes in all data in $\mathcal{T}$, whereas AutoCoT first performs clustering and only performs forward passes in the centroid queries in the best case. However, the difference is marginal because 1) the cost of Stage 1 is already very modest since $|\mathcal{T}|$ is small: if we further use batching as discussed previously, the difference would be between 1 forward pass (AutoCoT) to 2 (USP) if $|\mathcal{T}| = 64$ and batch size is 32; and 2) AutoCoT employes filtering process and it is possible that some LLM responses to the centroid queries do not meet the filtering criteria and repeated sampling is required, so in the average case, AutoCoT also needs to perform more LLM queries before a satisfactory pool of responses is built. In either case, we argue that the stronger empirical performance of USP justifies any slightly increased cost.
>
> - *Random Demo*: USP features an identical computational cost, as the key difference is the way demos are selected in Stage 1.
>
> > Lack of Comparative Analysis: While the paper shows that the USP method outperforms standard zero-shot baselines, it might not provide a detailed comparison with other state-of-the-art methods. This could make it difficult to assess the relative performance of the USP method.
>
> > Question 1B: Could you provide a more detailed comparison of the USP method with other state-of-the-art methods? It would be helpful to understand how USP performs in comparison, both in terms of accuracy and computational efficiency.
>
> We would like to stress that *comparison against SoTA methods is already included*.  We compared against standard zero-shot *and few-shot* (even though our method is zero-shot), and also generalized SoTA methods that previously did not consider a setup as broad as ours, like AutoCoT [1], for comparison in *all our experiments*. We also compared Random Demo, which is the closest possible adaptation of some of the competing methods that do not operate under an identical setup to ours. We discuss these decisions in Sec 4 Related Works on Page 6, and experimentally, we outperform *all of them*.
>
> The comparison in terms of computational cost can be seen in the response to Question 1A.
>
> > Question 1C: The paper mentions that the USP method requires only a small amount of unlabeled data. Could you provide more specifics on the amount of data used in your experiments and how this impacts the performance of the model?
>
> We stated on Page 4, Line 241, that we used 64 unlabeled samples per task (also Line 488, Line 503, and Line 526, Page 7). In Table 5 on Page 14, we further include the fraction of unlabeled samples to the entire test set as a demonstration of computational cost.
>
> Regarding the performance impact, we argue that requiring 64 unlabeled samples per task is already very modest. Given that unlabeled samples are almost always cheap to obtain either beforehand or on the fly, we did not explore an even lower value. We will include such an analysis in the camera-ready. USP will be at least as good or better when we have more data than the results reported in the paper, which already outperforms all competitive baselines.
>
> We also show in the paragraph “Few-shot USP” in Line 515, Page 8 and Appendix D that USP further benefits when we have some *labeled* data.
>
> > Question 1D: Are there any limitations or potential drawbacks to the USP method that were not discussed in the paper? If so, could you elaborate on these and how they might be addressed in future work?
>
> We included a limitation section on Page 9, Limitations. We’d be grateful if the reviewer could point out what, in their opinion, is lacking in the included section and what additional discussions are required?
>
> > Question 1E: Could you elaborate on the scalability of the USP method? Specifically, how does the performance of the method change with the size of the language model and the amount of available data?
>
> We discussed the potential effect of model size in the paragraph starting from Line 492, Page 7 (Discussion of main results). Namely, in Line 504, we stated, “Generally, we find the gain margin to be larger in generative tasks and larger and/or more advanced models. …”. This is empirically supported by experiments in two PaLM models of different sizes (62B and 540B). The results for PaLM-62B are included in Appendix D.1.
>
> The aspect of the impact of data has been answered in our response to Question 1C.
>
> > Question 1F: Are there any specific types of tasks or domains where the USP method performs particularly well or poorly? If so, could you provide some examples and explain why this might be the case?
>
> We included a section starting from Line 546, Page 8, “When does USP perform better?” that discusses which tasks USP performs better or worse, where we found the Stage 1 USP score across the unlabeled dataset, which measures the general degree of uncertainty and confidence of the LLM of a task, to be a reliable indicator on how much USP can help. We also discussed the influence of models and working mechanisms in the preceding paragraph, “How does USP work,” starting in Line 515, Page 7.
>
> > Clarity of Methodology: The explanation of the Universal Self-Adaptive Prompting (USP) methodology could be clearer. More details and perhaps a step-by-step walkthrough of the approach would be helpful.
>
> We thank the suggestion, but a step-by-step walkthrough is already included in a pseudo-code format in Algorithm 1, Page 4, and an illustrative figure of the pipeline is included in Fig 2, Page 2. We will be more than happy to include any further improvements as per the reviewer’s suggestion.
>
> > Comparisons with Other Methods: The paper could benefit from a more detailed comparison with existing methods. This could include a table summarizing the main differences and similarities, as well as a discussion of the advantages and limitations of each method.
>
> We thank the reviewer’s suggestion and will endeavor to include such a table in the final manuscript, which we agree might lead to better clarity. Nonetheless, independent of the form of presentation, the information requested here is already presented in the paper. For example, we include a thorough discussion of related works in Section 4 that compare and contrast our method with prior works.
>
> > Use of Figures: If not already included, the use of illustrative figures to explain the USP method could improve the readability of the paper and help readers understand the approach better.
>
> We kindly point the reviewer to Fig 2d, Page 2 for the illustrative figure requested – We’d be grateful if the reviewer could explain what else we can do to improve readability and flow.
>
> > Data Details: More information on the datasets used in the study would be helpful. This could include the size of the datasets, how they were collected, and any preprocessing steps.
>
> We kindly refer the reviewer to Appendix B.1 for dataset details and Appendix C.1 for preprocessing steps. We are happy to answer further questions if there’s anything else that needs to be added.
>
> > Discussion of Results: The paper could provide a more detailed discussion of the results, including potential reasons for the observed performance and any surprising or unexpected findings.
>
> We refer the reviewer for discussions of the main results and how and when USP works from Page 7 onwards. We’d be grateful if the reviewer could specify what they think is lacking in the already included analysis.
>
> > Conclusions and Future Work: The conclusion section could provide a more comprehensive summary of the main findings, their implications, and potential directions for future research.
>
> All requested information is in Conclusion and Limitations. If the reviewer has further suggestions, we’d be happy to include them in the final version of the manuscript.
>
> ### References
> [1] Zhang, Z., Zhang, A., Li, M., & Smola, A. Automatic chain of thought prompting in large language models. ICLR 2023.

---

### Official Review · Reviewer_4kJc · 2023-08-11

**Soundness:** 3

**Excitement:**

4: Strong: This paper deepens the understanding of some phenomenon or lowers the barriers to an existing research direction.

**Paper Topic And Main Contributions:**

This paper focuses on improving prompting and few-shot learning for large language models (LLMs) in zero-shot settings where no labeled data is available. The key problem it addresses is that existing prompting methods perform poorly in zero-shot scenarios due to the lack of guidance and inability to generate high-quality prompts without labels.

The main contributions are:

- Proposing a new prompting approach called Universal Self-Adaptive Prompting (USP) specifically designed for zero-shot learning.

- USP automatically generates pseudo-demonstrations from LLM predictions on unlabeled data and selects high-confidence examples using scoring functions tailored for different task types.

- Demonstrating that prepending these pseudo-demonstrations significantly improves zero-shot performance across 40+ NLU, NLG and reasoning tasks.

- Showing gains over standard zero-shot prompting and other zero-shot baselines, and competitive performance to few-shot learning with USP using only 64 unlabeled samples.

- Providing analysis on the impact of pseudo-demo quality and task-specific selectivity on the improvements from USP.

**Reasons To Accept:**

Some key strengths of this paper:

- Addresses an important open problem - improving zero-shot prompting for LLMs without labeled data.

- Proposes a novel method (USP) that is intuitive, simple, and achieves significant gains.

- Thorough empirical evaluation across 40+ diverse NLP tasks highlighting generalizability.

- Strong technical quality - rigorous experimental design, analysis of results, open-sourced code/models.

- State-of-the-art zero-shot results competitive with or superior to few-shot learning in many cases.

The main benefits to the NLP community:

- Provides an effective new technique to reduce labeled data needs when applying LLMs.

- Opens up new research directions for optimized zero-shot prompting.

- Yields insights into importance of high-quality demonstrations for LLMs.

- Improves availability of strong zero-shot prompting baselines for future research.

- Enhances capabilities of LLMs in low-resource scenarios and novel unseen tasks.

**Reasons To Reject:**

Some potential weaknesses of this paper:

- The approach relies on unlabeled data which may not always be readily available.

- More analysis could be provided on computational overhead of USP.

- Additional ablation studies validating design choices would strengthen the method.


The main risks if this paper was accepted:

- Reviewers may want to see experiments on a wider variety of languages and tasks.

- More work may be needed to demonstrate computational efficiency.

- The techniques used for scoring pseudo-demos could be critiqued as ad-hoc or lacking in sophistication.

**Reproducibility:**

4: Could mostly reproduce the results, but there may be some variation because of sample variance or minor variations in their interpretation of the protocol or method.

**Reviewer Confidence:**

3: Pretty sure, but there's a chance I missed something. Although I have a good feel for this area in general, I did not carefully check the paper's details, e.g., the math, experimental design, or novelty.

---

> ### Author Rebuttal · Authors · 2023-08-26
>
> We thank the reviewer for their constructive and positive feedback. Please see below for our response to the specific questions and concerns the reviewer had:
>
> > The approach relies on unlabeled data which may not always be readily available.
>
> While it is indeed possible that unlabeled data might not always be available, we argue that
> 1. Unlabeled data are much more readily available than labeled data, which standard in-context learning requires, especially when solving the task is challenging or laborious (e.g., for a summarization task, long articles are almost available anywhere, but high-quality summaries to be used as labeled demos are far less accessible). While standard zero-shot requires neither labeled nor unlabeled data, USP significantly outperforms it, which justifies the slightly higher requirement (that some unlabeled data is needed).
> 2. In real life, even if we do not have unlabeled data *to start with*, it is much easier to cache and store unlabeled data for a deployed model in an online fashion at almost no cost for later usage.
>
> > More analysis could be provided on computational overhead of USP.
>
> > More work may be needed to demonstrate computational efficiency.
>
> Computational costs: Computing the USP scores itself is cheap (can be done in any modern PC/laptop with CPU only), and the cost is thus bottlenecked by the amount of processing from the LLM side. In particular, the additional costs are:
> - Stage 1: Since we use 64 unlabeled samples, we only require 64 additional model queries for the CLS task and 64m (we use m = 6 in paper) for SFG and LFG tasks – note that we can also use batching to parallelize this step. The additional cost can be captured by the $|\mathcal{D}|\/|\mathcal{T}|$ (the fraction of the unlabeled samples to the size of the entire test set) column in Table 5 in supplementary cost, which as we can see is almost always negligible compared to the cost we need to incur anyway (except for some very small-scale toy tasks with small test tasks)
> - Stage 2: This stage is completely identical to standard few-shot in-context learning.
>
> Thus, compared to standard zero-shot learning, USP requires the additional Stage 1, which typically only adds a small amount of cost, as discussed above. In Stage 2, the LLM needs to process a longer context due to the use of pseudo-demos for in-context learning. However, this is due to the use of in-context learning and is not an additional cost uniquely attributable to USP – it is true for all other methods relying on ICL.
>
> Compared to few-shot learning, the only additional overhead is the use of Stage 1, but crucially, no labeled data is required at any point in time.
>
> > Additional ablation studies validating design choices would strengthen the method.
>
> We thank the reviewer’s suggestion. The primary ablation we have already included is the Random Demo baseline, which, as discussed in Sec 4 (Related works), serves both as a baseline that imitates various competing methods and as an ablation method since the primary contribution of USP is to use the scores provided in Sec 3. The results convincingly show that using the USP procedure is consistently stronger. The empirical analysis of several other design decisions, such as the use of diversity encouragement and its weight, was already included in COSP, the previous work we generalize from.
>
> For the final manuscript, we will follow the reviewer’s advice to include additional ablation studies, such as the effect of the size of $\mathcal{T}$ and the effect of some hyperparameters. It is worth noting, however, that we used a single set of hyperparameter values for all experiments rather than tuning them per task, which should give some evidence of the robustness of our method – this is also particularly critical in our setup, as we assume validation data, which is typically required for hyperparameter tuning, would not exist.
>
> > Reviewers may want to see experiments on a wider variety of languages and tasks.
>
> We indeed agree that it is always helpful to include more tasks. However, we argue that the tasks we consider are already very expansive (>40 tasks in total) and varied (NLU, commonsense reasoning, arithmetic, summarization, QA, etc.), and, to our knowledge, more so than most published papers and the experimental results consistently support our claim.
>
> We thank the reviewer’s suggestion and indeed plan to test on additional languages, which we defer to future work. It is worth noting, though, that nothing in the method is language-specific, and thus, we expect USP to work in a language-agnostic manner.
>
> > The techniques used for scoring pseudo-demos could be critiqued as ad-hoc or lacking in sophistication.
>
> We thank the reviewer for bringing up this point, and we agree that additional theoretical analysis would be helpful. However, we argue USP, by using well-studied concepts in uncertainty estimation, leads to a simple yet effective scoring function. Given that uncertainty calibration in LLMs is still a nascent area with many open research questions, we definitely agree that our work will benefit from further theoretical advancements in this topic.

---

### Meta-Review · Area_Chair_wQsR · 2023-09-17

**Recommendation:** 5

**Metareview:**

The work proposes a universal prompting approach for zero-shot inference by leveraging unlabeled data to construct in-context learning examples.

**Pro**:
- The reviewers all agree that improving zero-shot prompting is an important research topic, especially in the case where labeled data is not available.
- The authors conduct a large suite of experiments and show that the proposed approach consistently improves model performance by a large margin across various types of tasks. The evaluation and analysis are very through and exhaustive.
- The paper is well written and easy to follow.

**Con**:
- Some reviewers concern that paper only conducts experiments on close-sourced models and it’s not clear if it transfers to open-sourced models.

---

### Decision · Program_Chairs · 2023-10-07

**Decision:**

Accept-Main

**Comment:**

The work proposes a universal prompting approach for zero-shot inference by leveraging unlabeled data to construct in-context learning examples.

**Pro**:
- The reviewers all agree that improving zero-shot prompting is an important research topic, especially in the case where labeled data is not available.
- The authors conduct a large suite of experiments and show that the proposed approach consistently improves model performance by a large margin across various types of tasks. The evaluation and analysis are very through and exhaustive.
- The paper is well written and easy to follow.

**Con**:
- Some reviewers concern that paper only conducts experiments on close-sourced models and it’s not clear if it transfers to open-sourced models.